# *In vivo* Dicer-2 interactome during viral infection reveals novel pro and antiviral factors in *Drosophila melanogaster*

Claire Rousseau[1¤], Thomas Morand[1], Gabrielle Haas[1], Emilie Lauret[1], Lauriane Kuhn[2], Johana Chicher[2], Philippe Hammann[2], Carine Meignin[1]*

**1** Université de Strasbourg, M3i CNRS UPR9022, Strasbourg, France, **2** Plateforme Protéomique Strasbourg-Esplanade, Université de Strasbourg, CNRS UAR1589, Strasbourg, France

¤ Current address: Institute for Virology and Immunobiology, Julius-Maximilians-Universität-Würzburg, Würzburg, Germany
* c.meignin@ibmc-cnrs.unistra.fr

## Abstract

RNA interference has a major role in the control of viral infection in insects. It is initialized by the sensing of double stranded RNA (dsRNA) by the RNAse III enzyme Dicer-2. Many *in vitro* studies have helped understand how Dicer-2 discriminates between different dsRNA substrate termini, however it is unclear whether the same mechanisms are at work *in vivo*, and notably during recognition of viral dsRNA. Indeed, although Dicer-2 associates with several dsRNA-binding proteins (dsRBPs) that can modify its specificity for a substrate, it remains unknown how Dicer-2 is able to recognize the protected termini of viral dsRNAs. In order to study how the ribonucleoprotein network of Dicer-2 impacts antiviral immunity, we used an IP-MS approach to identify *in vivo* interactants of different versions of GFP::Dicer-2 in transgenic lines. We provide a global overview of the partners of Dicer-2 *in vivo*, and reveal how this interactome is modulated by different factors such as viral infection and/or different point mutations inactivating the helicase or RNase III domains of GFP::Dicer-2. Our analysis uncovers several previously unknown Dicer-2 interactants associated with RNA granules, i.e., Me31B, Rump, eIF4E1, eIF4G1, Rin and Syncrip. Functional characterization of the candidates, both in cells and *in vivo*, reveals pro- and antiviral factors in the context of an infection by the picorna-like DCV virus. This work highlights protein complexes assembled around Dicer-2 *in vivo,* and provides a resource to investigate their contribution to antiviral RNAi and related pathways.

## Author summary

Invertebrates rely mainly on RNA interference (RNAi) as a key defense mechanism against viral infections. The central protein of this process is an enzyme called Dicer-2, which senses and processes viral double-stranded RNA (dsRNA),

**Data availability statement:** The mass spectrometric data were deposited to the ProteomeXchange Consortium via the PRIDE partner repository at https://www.ebi.ac.uk/pride/archive/projects/PXD038898 Script used for data analysis on FigShare at https://doi.org/10.6084/m9.figshare.27909918.v4

**Funding:** This work of the Interdisciplinary Thematic Institute IMCBio, as part of the ITI 2021-2028 program of the University of Strasbourg, CNRS and Inserm, was supported by IdEx Unistra (ANR-10-IDEX-0002), EUR IMCBio (ANR-17-EURE-0023) and SFRI-STRAT'US project (ANR 20-SFRI-0012) and within the framework of the French 2030 to C.M.. T.M was supported by fellowship from EUR IMCBio (ANR-17-EURE-0023). Part of this work was also funded by the ViroMOD project, under the framework of the Regional Cooperation Fund for Research (FRCR) to C.M.. The funders had no role in study design, data collection and analysis, decision to publish, or preparation of the manuscript.

**Competing interests:** The authors have declared that no competing interests exist.

a common signature of viruses, to inhibit infection. While the general mechanism of RNAi is well described, some key aspects remain unknown, such as how Dicer-2 recognizes the ends of viral dsRNA, which are often shielded by protective structures.

To explore this, we studied how Dicer-2 interacts with other proteins in whole fruit flies (*Drosophila melanogaster*). We identified several proteins associated with Dicer-2 under different conditions, such as during viral infection. Many of the identified proteins, including Me31B, Rump, eIF4E1, eIF4G1, Rin and Syncrip, are linked to specialized structures involved in RNA regulation. We also show that some of these proteins also have an impact on viral infection.

Our findings provide insights into the network of proteins that interact with Dicer-2 in the whole organism, and how this network is altered by viral infection. This work provides a valuable resource for understanding antiviral immunity and paves the way for further research into RNAi-related pathways in insects and beyond.

## Introduction

In metazoans, virus-derived double-stranded RNAs (dsRNAs) allow the detection of a broad range of viruses by pattern recognition receptors (PRRs), leading to the activation of antiviral innate immunity [1]. Dicer proteins are dsRNA sensors with an endoribonuclease activity from the RNase III family, enabling the production of microRNAs (miRNAs) and small interfering RNAs (siRNAs) [2]. *Drosophila melanogaster*, like other arthropods, encodes two Dicer proteins: Dicer-1, dedicated to miRNA processing, and Dicer-2 that is required for siRNA biogenesis [3]. Insects largely rely on the siRNA pathway for antiviral defense through the detection of viral dsRNA by Dicer-2 [4–9]. The virus-derived siRNAs (vsiRNAs) produced by Dicer-2 are loaded onto the protein Argonaute2 (AGO2) to form the RNA-induced silencing complex (RISC), which targets and degrades complementary RNAs [5,9,10]. Dicer-2 has also been proposed to participate in the regulation of the expression of antiviral genes, in addition to its function in RNAi [11,12].

Dicer-2 is able to recognize and discriminate between two types of dsRNA termini and subsequently initiates two distinct types of cleavage mechanisms, called processive and distributive dicing [13–16]. *In vitro*, blunt dsRNA promotes processive cleavage, whereby the helicase domain of Dicer-2 binds the dsRNA termini and thread through the dsRNA molecule using ATP hydrolysis to produce multiple siRNA duplexes in one go. In contrast, dsRNA with a 3'overhang promotes distributive cleavage, whereby the 5′-monophosphate of the dsRNA substrate is anchored by the phosphate-binding pocket in the Dicer-2 Platform•PAZ domain [17,18] and Dicer-2 dissociates after each high-fidelity cleavage in an ATP-independent manner [19].

The helicase domain of all Dicer proteins is phylogenetically related to the retinoic acid-inducible gene-I (RIG-I)-like receptors RLRs [2,11]. RLRs are cytosolic dsRNA sensors that induce an interferon response in vertebrates, suggesting that the helicase domain is a sensor of viral infection. Viral dsRNAs can be synthetized as an intermediate product of viral replication in RNA viruses or can originate from the convergent transcription of DNA viruses. In drosophila, all viruses tested so far are detected by Dicer-2 and induce the production of vsiRNAs [5,6,8,9,20]. However, as the dicing mechanism relies on the recognition of the termini of the substrate, and viral dsRNAs produced during viral replication *in vivo* do not usually contain free ends, this raises the question of how Dicer-2 and RLRs are able to sense viral infection. For example, the genome of the picorna-like *Drosophila C virus* (DCV, *Dicistroviridae*) is protected at the 5'end by a covalently bound protein called VPg and at the 3'end by a polyA tail.

To perform its function, Dicer-2 associates with several dsRNA-binding proteins (dsRBPs). The two accessory dsRBPs Loqs and R2D2 bind the helicase domain of Dicer-2, consisting of 3 subdomains, Hel1, Hel2i and Hel2 (**Fig 1A**) [21–23]. More specifically, Loqs binds the Hel2 domain [24] and R2D2 binds the Hel2i domain [23]. Other dsRBPs (e.g., TRBP, PACT, PKR and ADAR) also interact with the helicase domain of human Dicer [25–28]. The helicase domain therefore appears to be of utmost importance for the interaction of Dicer-2 with different regulatory proteins. Moreover, the helicase domain of human Dicer and drosophila Dicer-2 is important for antiviral response [11,19,28–31].

In drosophila, Dicer-2 forms a heterodimer with the dsRBP R2D2, and this association is essential for several steps in the siRNA biogenesis. Although Dicer-2 is able to cleave pre-miRNAs *in vitro*, R2D2 and inorganic phosphate restrict the specificity of Dicer-2, preventing pre-miRNA cleavage [14,32]. R2D2 further amplifies this distinction between Dicer-1 and Dicer-2 by localizing Dicer-2 to cytoplasmic foci, called D2 bodies, where endo-siRNAs will be cleaved, away from the pre-miRNAs [22]. After siRNA processing by Dicer-2, R2D2 is also required for exo-siRNA loading onto AGO2 [33–35], which is then stabilized by the Hsc70/Hsp90 chaperone machinery [36–38]. One strand of the siRNA duplex is then discarded, such that the remaining strand can guide the RISC complex to the complementary target RNA for silencing. For this step, R2D2 is needed again, as it functions as a protein sensor for thermodynamic differences in the base-pairing stabilities of the 5'end of the siRNAs. Thus, it allows the siRNA loading onto AGO2 in a specific orientation, thereby determining which strand will be discarded and which one will serve as the guide [23,39].

Another dsRBP, the TRBP drosophila homologue Loquacious (Loqs), plays an important role in the determination of Dicer-2 specificity. Because of alternative splicing, there are four distinct Loqs isoforms, with specific activities in the Dicer-1-dependent miRNA biogenesis pathway or the Dicer-2-dependent endo-siRNA pathway. While the function of Loqs-PC remains unknown, Loqs-PA and Loqs-PB interact with Dicer-1 for miRNA biogenesis, and Loqs-PD interacts with Dicer-2 for the biogenesis of endo-siRNAs [35,40–44]. Intriguingly however, Loqs-PD is not required for the targeting of viral dsRNA for the viruses tested [29]. This last isoform is able to modulate the termini dependence of Dicer-2 by enabling the cleavage of sub-optimal substrates such as dsRNA with blocked, structured, or frayed ends [15]. This modulation is not achieved by changing the cleavage mode (i.e., processive and distributive), but rather by affecting the probability for Dicer-2 to cleave the sub-optimal substrate [45].

To unravel the protein network associated with Dicer-2 during viral infection and test the impact of Dicer-2 partners in antiviral immunity, we used an interactomics approach through immunoprecipitation followed by mass spectrometry (IP-MS), in a fly line expressing the wild-type version of Dicer-2 fused to a GFP tag, GFP::Dicer-2^WT. To stabilize interactions between Dicer-2 and the dsRNA, and thus potentially identify interactants that play a role in the viral dsRNA sensing by Dicer-2, we also used two Dicer-2 mutants able to sense dsRNA but unable to process it. One of those mutants expresses a GFP-tagged version of Dicer-2 with the G31R mutation on the Hel1 domain, called GFP::Dicer-2^Hel1. This Dicer-2 mutant processes 3'overhang dsRNA substrates using distributive dicing, but not blunt dsRNA substrates, which requires ATP hydrolysis for processive dicing. It has been described in previously published work, together with the impact of this mutation on endo- and exo-siRNA production [19]. We also used another Dicer-2 mutant fly line, GFP::Dicer-2^RNaseIII, with two mutations in the RNase III domains, in the hope of identifying more transient interactions,

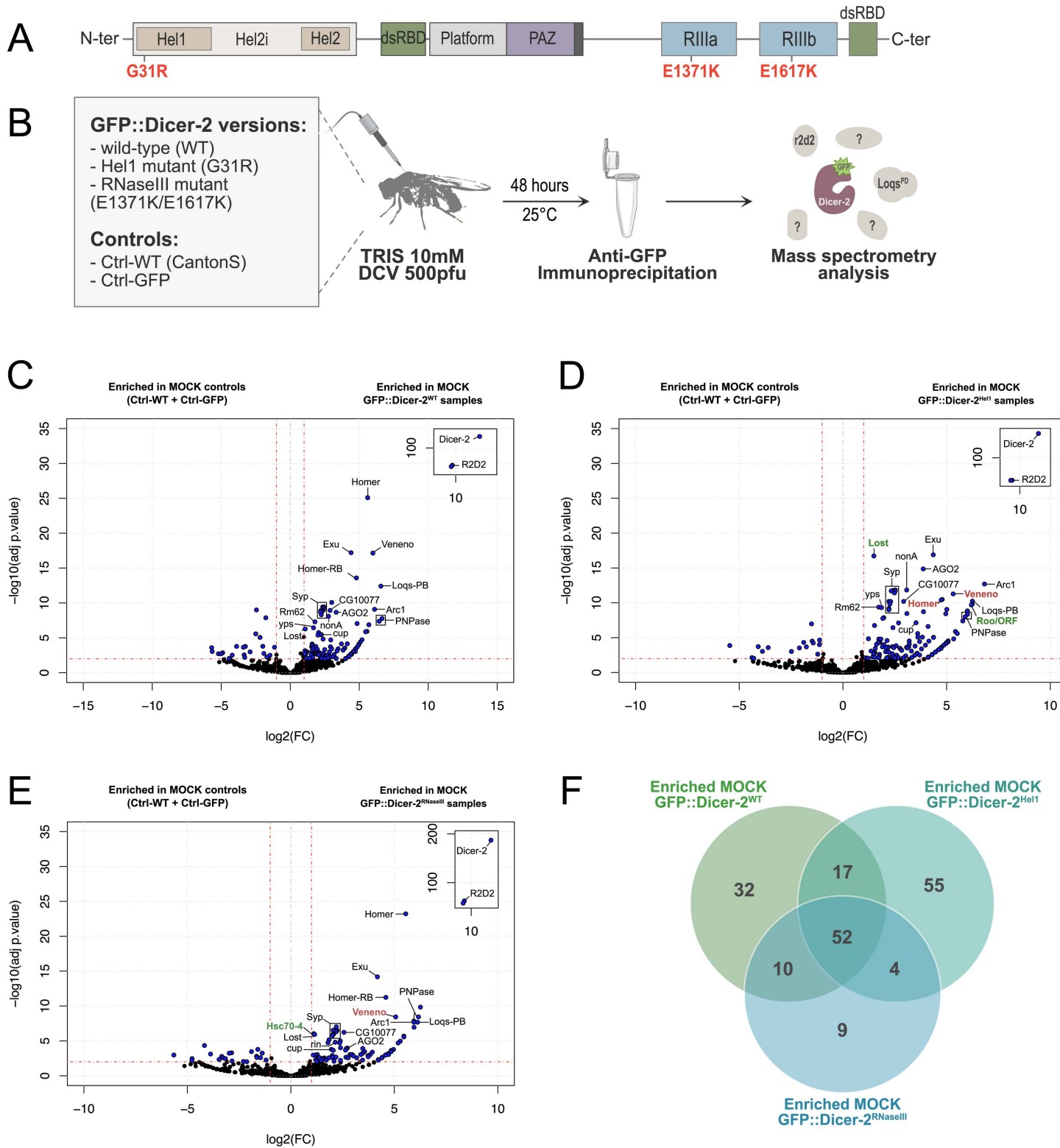

**Fig 1. Impact of the Dicer-2 mutations on the RNP network.** (A) Schematic representation of Dicer-2 domain architecture. The Dicer-2 protein (1722 amino acids) is composed of a helicase domain (which contains the Hel1, Hel2i and Hel2 sub-domains), two double-strand RNA binding domains (dsRBD), a PAZ domain, and two ribonuclease III domains (RIIIa and RIIIb). G31R and E1371K/E1671K represent the point mutations of *dicer-2* used

in this study. (B) Scheme illustrating the experimental strategy for IP & LC-MS/MS used to identify Dicer-2 partners *in vivo* in mock-infected (TRIS) and DCV-infected conditions. All genotypes used are presented in the grey box. This experiment has been performed in triplicate. Image sourced from smart. servier.com (https://smart.servier.com/smart_image/smart-microtube/), licensed under CC BY 4.0. (C-E) Volcano plots representing the fold changes and adjusted *p*-value (adj*p*) of the Dicer-2 partners in GFP::Dicer-2^WT (C), GFP::Dicer-2^Hel1 (D) and GFP::Dicer-2^RNaseIII (E) lines in mock samples (TRIS injection) *versus* the control lines (Ctrl-WT + Ctrl-GFP). Fold-changes and *p*-values were obtained using a negative binomial test and *p*-values were corrected by the Benjamini-Hochberg method to obtain adjusted *p*-values. All the proteins with fold change > 2 and an adjusted *p*-value < 0.01 are represented in blue. Green labels correspond to proteins that were more enriched in the mutant than in GFP::Dicer-2^WT, and red labels correspond to proteins that were less enriched in the mutant than in GFP::Dicer-2^WT. (F) Venn diagram showing the number of proteins identified in each GFP::Dicer-2 line. After removing duplicate matches to a single protein, candidates were selected with a fold-change > 2 and an adjusted *p*-value < 0.01. See S1 Text.

as this mutant is able to bind dsRNA but not to cleave it. We report the identification and functional characterization of novel Dicer-2 interactants. Among them, we observed proteins such as Me31B and eIF4E1, for which the interaction with Dicer-2 is RNA-independent, whereas the protein Syncrip (Syp) interacts with Dicer-2 in an RNA-dependent manner. By performing two RNAi screens, both in S2 cell culture and *in vivo*, we demonstrate that several of those proteins have an impact on viral DCV infection, such as the protein Rasputin (Rin). This result highlights a panel of interaction profiles assembled around Dicer-2 *in vivo,* and provides a resource to investigate the contribution of the different interactants to antiviral RNAi and related pathways.

## Results

### Definition of the Dicer-2 interactome *in vivo*

To study the dynamics of the protein network surrounding Dicer-2 *in vivo* in response to viral infection, we complemented *dicer-2* null mutant fly lines with WT or mutant versions of GFP::Dicer-2 and injected them with either TRIS (mock-infection) or DCV. After immunoprecipitation of the different GFP::Dicer-2 versions, their protein partners were then identified by LC-MS/MS (Figs 1A, 1B, S1A and S1B). The Hel1 and RNase III mutants should provide an overview of the interactome of Dicer-2 while its activity is reduced, but its interaction with dsRNA is increased. Both mutations are expected to increase the proportion of Dicer-2 proteins bound to dsRNA, due to slower or impaired processing. All experiments were performed in adult flies, and the ability of the wild-type version of Dicer-2 (GFP::Dicer-2^WT) to rescue the *dicer-2* null mutation was demonstrated in previous studies [6,19,46]. The impact of the mutations on RNAi efficiency can be observed in S1B Fig, as the flies contain the *w^IR* transgene that alters their eye color from red (indicating an impaired RNAi pathway) to white when RNAi is working properly. Furthermore, two control fly lines were used to determine non-specific interactants, both expressing endogenous *dicer-2* normally: a wild-type CantonS line (Ctrl-WT) and a transgenic line expressing GFP ubiquitously (Ctrl-GFP). Of note, the level of expression of GFP::Dicer-2 in the complemented lines is comparable to the endogenous expression of Dicer-2 in control lines (S1A Fig). In total, 2509 protein hits were identified by LC-MS/MS across all samples.

### Comparative analysis of the interaction profiles of Dicer-2^WT and mutants

We began our study by examining the respective impacts of the Dicer-2 Hel1 and RNase III domain mutations under uninfected conditions, focusing initially on the mock samples. Three separate statistical analyses were performed to compare the interactomes of each GFP::Dicer-2 line to each other (Fig 1C-E). For each analysis, samples from the two control lines (Ctrl-WT and Ctrl-GFP) were grouped together as a single control to facilitate comparison with the GFP::Dicer-2 lines, as they clustered together in a Multidimensional Scaling (MDS) analysis (S1C Fig), independently of the GFP::Dicer-2 lines. We then used a negative-binomial test to identify proteins enriched in each GFP::Dicer-2 line compared to the control lines (Ctrl-WT + Ctrl-GFP), with a fold-change > 2 and an adjusted *p*-value < 0.01. This then allowed us to group the different

proteins interacting with Dicer-2 into categories depending on their enrichment in the different GFP::Dicer-2 lines (Figs 1F and S2).

We can first observe that 52 proteins appear to be enriched in all three GFP-Dicer-2 lines (Fig 1F), suggesting very strong interactions with Dicer-2. As expected, one of those 52 proteins, R2D2, was always associated with the greatest Log2(FC) and lowest p-value for all three GFP-Dicer-2 lines, consistent with the presence of a stable heterodimer Dicer-2/R2D2 *in vivo* [33]. Moreover, several other known interactants of Dicer-2 were highlighted amongst the most enriched proteins for all GFP-Dicer-2 lines, such as Loqs and AGO2 (Figs 1C-F and S2). Among the most enriched proteins in all GFP::Dicer-2 samples are also the proteins Homer, Veneno, and Exuperantia (Exu). Although the Tudor protein Veneno has been shown to be involved in small RNA pathways [47,48], a truncated form of Veneno has also been demonstrated to exhibit antiviral activity against Drosophila A virus (DAV), independently of the RNAi pathway [49]. Homer and Exu are mostly known for their roles in sleep/locomotion and the establishment of oocyte polarization, respectively [50,51]. Finally, among the top candidates in this category were the RNA-binding protein Syncrip (Syp), the RNA exonuclease Polynucle-otide Phosphorylase (PNPase), and the Gag-like retrotransposon Arc1 [52–56].

These analyses also highlight differences in the interaction profiles of the Hel1 and RNase III mutants compared to GFP::Dicer-2$^{WT}$. We can indeed observe the absence of some proteins in the interaction profiles of the Hel1 mutant (e.g., DIP1), the RNase III mutant (e.g., Rm62, Rump, Me31B) or both (e.g., Aats-glupro, CkIα), suggesting that these domains or the full Dicer-2 activity might be necessary for the interactions between Dicer-2 and those proteins. Moreover, some of the proteins enriched in all three GFP::Dicer-2 lines seem to have a lower p-value in the GFP::Dicer-2 mutant interactomes compared to the WT (e.g., Veneno and Homer, Fig 1C-E), suggesting that these interactions might also be impacted to some extent by the mutations.

Finally, we can also observe proteins that are enriched only in one or both of the mutants but not in the GFP::Dicer-2$^{WT}$ line. This could suggest weaker interactions, stabilized either by the slower processing of the GFP::Dicer-2$^{Hel1}$ mutant or by the inability of the GFP::Dicer-2$^{RNaseIII}$ mutant to cleave RNA, both expected to result in a longer interaction of Dicer-2 with the RNA. Of note, we can observe that although Lost was enriched in all three GFP::Dicer-2 lines, the statistical significance of this interaction is higher in the GFP::Dicer-2$^{Hel1}$ mutant. Indeed, its ranks just after R2D2 and Exu in our analysis, with a –log10(adjp) value of 17, whereas it was not as highly ranked for the other two GFP::Dicer-2 lines. Moreover, we can also observe 4 proteins enriched specifically in the two mutant GFP::Dicer-2 lines, but not in the WT, amongst which Roo/ORF, which becomes one of the top interactions in the mutants.

Taken together, these results show that, although there is an important overlap between the interaction profiles of Dicer-2$^{WT}$ and its mutants, including known protein partners, the interaction profile of Dicer-2 is modulated by the Hel1 and RNase III mutations, potentially allowing the identification of new, less stable interactants of Dicer-2.

## DCV infection strongly alters the interactome of Dicer-2$^{WT}$ and its mutants

Next, we focused on the DCV-infected samples to analyze the impact of the DCV infection on the three Dicer-2 interaction profiles established above. Using the same methodology as before, we thus identified proteins enriched in each of the GFP::Dicer-2 lines after infection (Fig 2A-C), and separated them into the same categories depending on their enrichment in the different GFP::Dicer-2 lines (Fig 2D). By comparing the resulting volcano plots to the ones obtained with the mock samples, we can immediately observe that the DCV infection has a profound impact on the Dicer-2 protein network (e.g., proteins highlighted in purple in Fig 2A-C). As a general trend, we can observe that more proteins were significantly enriched in all categories in the DCV-infected samples compared to the mock-infected samples, and especially in the GFP::Dicer-2$^{Hel1}$ mutant.

By comparing the 52 proteins enriched in all three GFP::Dicer-2 lines under mock conditions with the 83 enriched in all three lines after DCV infection, we can have some insight on the interactions that seem very stable in both cases (S3A Fig). This reveals 56 proteins (e.g., FASN2, Nc73EF, α-KGDHC, NorpA, Aats-glupro, Dp1), represented in purple in

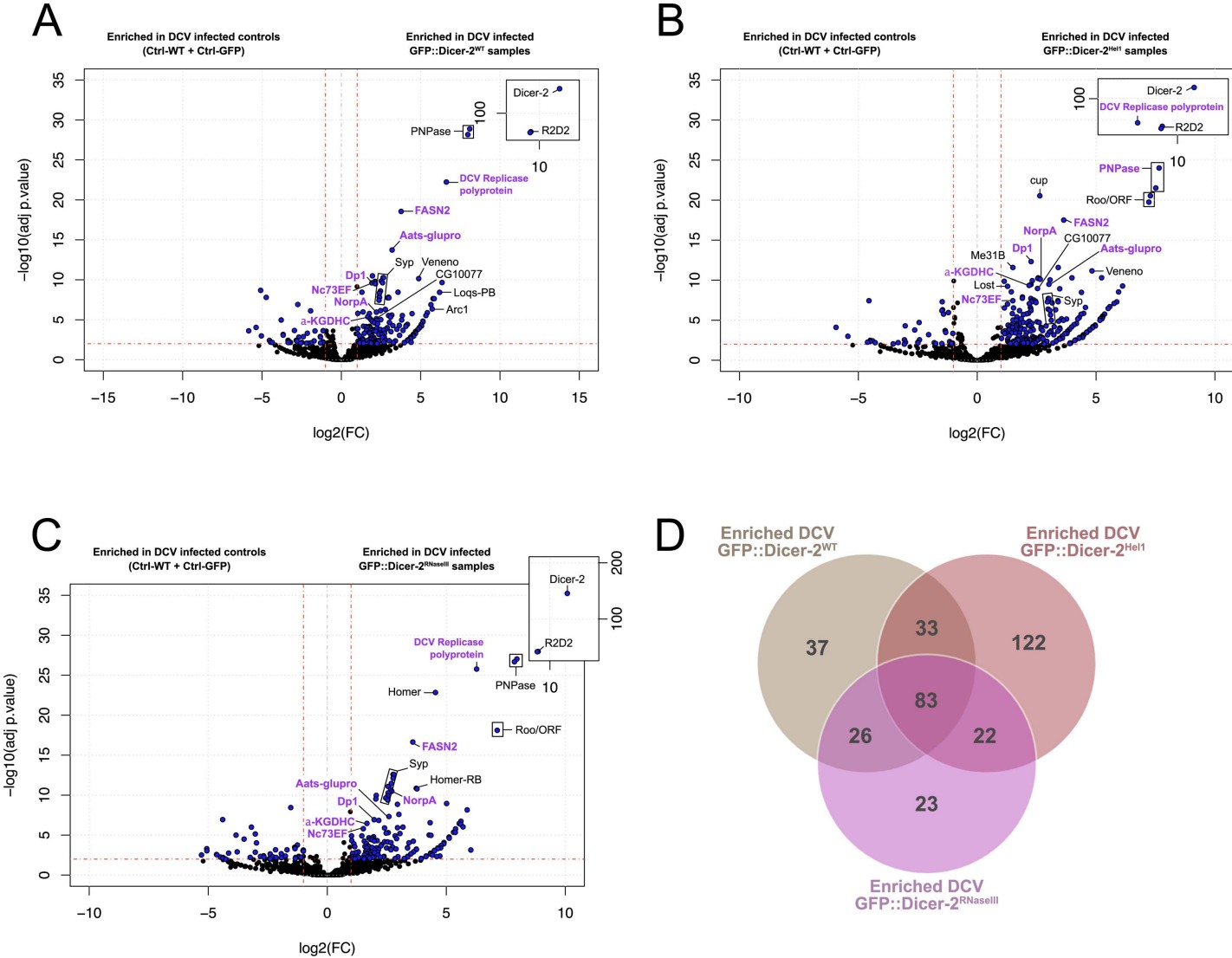

**Fig 2. Impact of the DCV infection the RNP network of Dicer-2.** (A, B, C) Volcano plots representing the fold changes and adjusted *p*-value (adjp) of the GFP::Dicer-2$^{WT}$ (A), GFP::Dicer-2$^{Hel1}$ (B) and GFP::Dicer-2$^{RNaseIII}$ (C) partners in DCV-infected adult flies *versus* the control lines (Ctrl-WT + Ctrl-GFP). Fold-changes and *p*-values were obtained using a negative binomial test and *p*-values were corrected by the Benjamini-Hochberg method to obtain adjusted *p*-values. All the proteins with fold change > 2 and an adjusted *p*-value < 0.01 are represented in blue. Names represented in purple correspond to proteins that were enriched in all three GFP::Dicer-2 lines in the DCV-infected samples, but not in all three lines in mock-infected samples (see S3 Fig). (D) Venn diagram showing the number of proteins identified in each GFP::Dicer-2 line in DCV-infected adult flies. After removing duplicate matches to a single protein, candidates were selected with a fold-change > 2 and an adjusted *p*-value < 0.01.

Fig 2A-C, that were enriched across all three lines following infection but were not enriched in at least one line under mock conditions (Figs 2A-C and S3). This might indicate interactions that are more stable upon infection and thus highlight proteins that have an impact on Dicer-2 during DCV infection (either pro- or antiviral). Of note, these interactants include the DCV Replicase polyprotein. This polyprotein contains several non-structural proteins, which are expressed as one polyprotein before cleavage and maturation and share an annotation. In our data, the peptides identified as the DCV Replicase polyprotein match almost exclusively the RNA-dependent RNA polymerase (RdRp) and the viral suppressor of RNAi DCV1A. This is most likely due to Dicer-2 being present on the same viral RNA molecule as the DCV RdRp and DCV1A.

Moreover, we can identify again the main proteins of the different GFP::Dicer-2 mock interaction profiles, PNPase, Veneno, Exu, Homer, Arc1, Loqs and Syp, as they are all part of the 27 proteins enriched in all three GFP::Dicer-2 lines for both mock and DCV-infected samples. However, their relative rankings seem to be disrupted by the infection, with PNPase becoming the second strongest interaction after R2D2 for two of the three lines, above both Veneno, Exu and Homer. This suggests that, unlike PNPase, these three proteins are more likely involved in functions of Dicer-2 unrelated to antiviral RNAi.

Finally, these analyses also show mutation-specific alterations of the interaction profiles of Dicer-2 after DCV infection (represented in green in Fig 2A-C). For example, Roo/ORF, still only enriched in both mutants but not Dicer-2 WT, seems to be more enriched in DCV-infected samples. In addition, Homer is more enriched in GFP::Dicer-2$^{RNaseIII}$ samples compared to the other two lines, which might be explained by Dicer-2 being unable to detach from the RNA upon infection, therefore lessening the alteration of its interaction profile by DCV infection.

## Visualization of the differential interaction profiles of Dicer-2 and generation of a candidate list

To validate the robustness of our mass spectrometry data, we then selected a representative subset of enriched proteins for further experimental validation. This was done by using the Significance Analysis of INTeractome (SAINT) express computational tool, as a complement to the analyses described above, to obtain an overview of the impact of both infection and the Hel1 and RNase III mutations on the differential interaction profiles of Dicer-2.

The SAINTexpress tool highlighted 288 proteins that were significantly enriched compared to the controls overall (S2 Table). We then selected the top 15% proteins of this analysis (44 candidates) to establish a list of candidates to validate further, which contains the main proteins highlighted by the previous analyses (Fig 3A). Indeed, this list of 44 interactants incorporates most of the main proteins highlighted by all three mock analyses (Figs 1C-E and S3B, e.g., R2D2, Veneno, Homer, Exu, Syp), and some that were significantly enriched in GFP::Dicer-2$^{WT}$ but not in one or both mutants (S2 Fig, e.g., Rm62, Lig), and also some that were only enriched in all three GFP::Dicer-2 lines upon infection (S3C Fig, e.g., FASN2, Dp1, alpha-KGDHC), but not in the three lines for mock-infected samples. This subset of candidates therefore contains proteins that fit the expected interaction profile of stables interactants of Dicer-2 under uninfected conditions, as well as those whose interactions are influenced by the mutations, or preferentially occur during infection. Visualization of this list of candidates using Prohits-viz [57] allowed us to recapitulate the impact of both the mutations and the infection (Fig 3A), and the predicted interactions between those candidates was represented using the STRING database (Fig 3B).

This list of candidates, which is not meant to be an exhaustive list, represents a subset of enriched proteins that we will use to provide proof of principle that the data are robust and biologically meaningful, providing the scientific community with a panel of candidates that can be explored based on their specific research interests.

## Validation of the interaction between Dicer-2 and new protein partners

The interactions between the candidates and GFP::Dicer-2 were then validated by immunoprecipitation and western blot analysis (Fig 4). Since these interactions were validated *in vivo*, we were unfortunately restricted to candidates for which antibodies were available for the endogenous protein. We were however still able to confirm the interaction of Me31B, Rump, eIF4E1 and Syp with GFP::Dicer-2$^{WT}$ as well as the two mutants (Fig 4A). These proteins are known to be part of the same RNA-protein complexes [52,58–60]. The interaction between Dicer-2 and its partners does not appear to be dependent on DCV infection (Fig 4B). On the contrary, after infection with DCV we can observe a weaker co-immunoprecipitation of the different candidates, which could suggest that Dicer-2 associates with mRNP complexes linked to other functions of Dicer-2, like translation repression. Of note, the interaction of Syp with Dicer-2 seems unstable, as the band was sometimes undetectable (S4A Fig). We also note that the interactions are often enriched with the GFP::Dicer-2$^{Hel1}$ mutant, which is consistent with the MS analysis, in which more spectral counts are observed with the GFP::Dicer-2$^{Hel1}$ line (Fig 4A). This could be due to a stabilization of these interactions in the GFP::Dicer-2$^{Hel1}$ mutant.

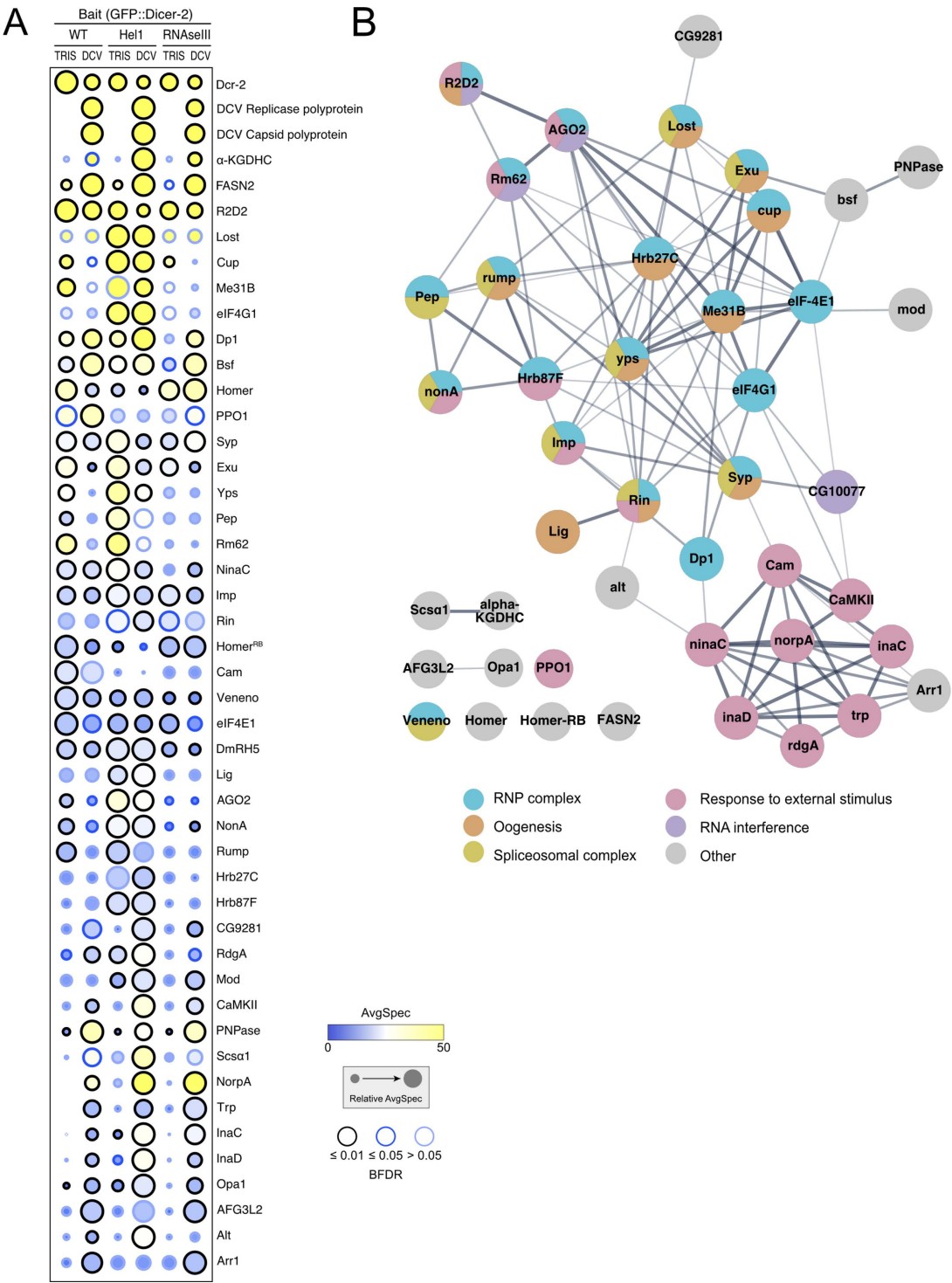

**Fig 3. Dicer-2 interactome network during viral infection *in vivo* to select a list of candidates.** (A) Top candidates of the SAINTexpress analysis from 3 independent experiments for each condition. The average number of spectra (AvgSpec), the relative abundance and Bayesian false discovery rate (BFDR) are represented. Among the top proteins, R2D2 is the second interactant of Dicer-2 for all three GFP::Dicer-2 lines and independent of the infection. (B) Global interaction network obtained with STRING/Cytoscape from the top 15% Dicer-2 partner proteins (i.e., top 44 proteins). The more an interaction is described in the literature, the thicker and darker the line connecting the two partners.

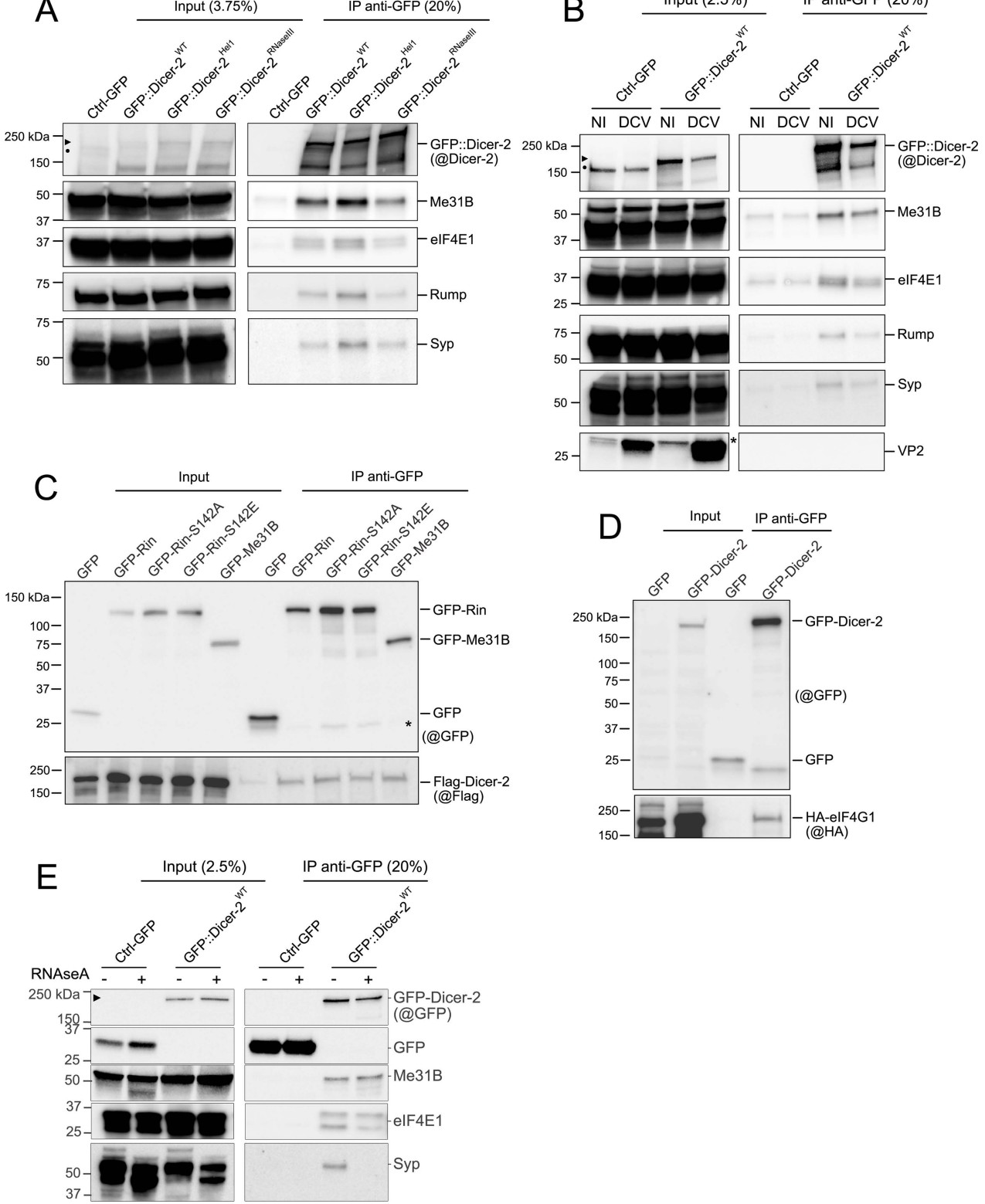

**Fig 4. Validation of the Dicer-2 interactome.** Results of the mass spectrometry analysis were confirmed by immunoprecipitation of GFP-tagged proteins in transgenic flies expressing GFP::Dicer-2 or in S2 cells. (A) Immunoprecipitation of GFP::Dicer-2^WT, GFP::Dicer-2^Hel1 and GFP::Dicer-2^RNaseIII transgenic flies in comparison to the Ctrl-GFP line. Representative result of n = 3 independent experiments. (B) Immunoprecipitation of GFP::Dicer-2^WT

flies in comparison to the Ctrl-GFP line in non-infected (NI) or DCV-infected conditions. Representative result of n = 3 independent experiments (C) Co-immunoprecipitation of GFP-Rin, GFP-Rin^S142A, GFP-Rin^S142E and GFP-Me31B with Flag-Dicer-2 ectopically expressed in S2 cells. Representative of n = 4 independent experiments. (D) Co-immunoprecipitation of GFP-Dicer-2 with HA-eIF4G1 ectopically expressed in S2 cells. Representative of n = 4 independent experiments. (E) Immunoprecipitation of GFP::Dicer-2^WT in comparison to the Ctrl-GFP line, after treatment (+) or not (-) with RNase A. Representative result of n = 3 independent experiments (►) Size of GFP::Dicer-2; (●) Size of endogenous Dicer-2; The asterisk (*) represent a non-specific band with anti-VP2 and anti-GFP.

Moreover, in the raw MS data very few spectra for Syp, Rump and eIF4E1 were identified with the Ctrl-GFP and Ctrl-WT lines, while more spectra were identified in the controls for Me31B (S1 Table). The fact that Me31B was, although in lower amounts than for the GFP::Dicer-2 lines, also found in the Ctrl-GFP IP (Fig 4B) is therefore consistent with the MS results. Moreover, these interactions do not seem to require DCV infection to occur, which we also observed in the MS data, as spectra for those proteins were also identified in the non-infected samples (Fig 4A and S1 Table).

To confirm additional candidates, we performed co-immunoprecipitation (co-IP) experiments in S2 cells with three candidates: Me31B (used as a positive control, as we had already confirmed this interaction), Rin, and eIF4G1. Dicer-2 and these three candidates are known to be components of different RNP complexes, including D2-bodies, P-bodies, stress granules, and complexes associated with translation [22,61–63]. Since the localization of Rin in stress granules is phosphorylation-dependent, we tested the interaction between Dicer-2 and Rin wild-type, a non-phosphorylatable mutant (S142A), and a phospho-mimic mutant (S142E) [62]. First, we confirmed that Flag-Dicer-2 co-immunoprecipitates with GFP-Me31B, as previously observed *in vivo* (Fig 4C). In S2 cells, Flag-Dicer-2 also co-immunoprecipitated with GFP-Rin, regardless of its phosphorylation status at serine position 142 (Fig 4C). The interaction between eIF4E1 and eIF4G1 has already been described as part of the translation initiation complex [63]. Since GFP::Dicer-2 interacts with eIF4E1 *in vivo*, we tested the interaction between Dicer-2 and eIF4G1 by co-IP in S2 cells. In this system, we demonstrated that HA-eIF4G1 co-immunoprecipitates with GFP-Dicer-2 (Fig 4D). These results suggest that Dicer-2 interacts with the translation initiation complex (Fig 4A and 4D).

In total, we were therefore able to confirm the interaction of Dicer-2 with six of the candidates highlighted by our MS experiment: three were validated with the endogenous proteins *in vivo* (eIF4E1, Rump and Syp), two in S2 cells (Rin and eIF4G1), and one was validated both *in vivo* and in cells (Me31B).

## RNA participates in the interaction between Dicer-2 and some of its associated proteins

As we decided to perform the immunoprecipitations in low stringency conditions, the IP-MS analysis allowed us to identify a high number of interactions with Dicer-2. Those could be direct protein-protein interactions, but also indirect interactions through other proteins or RNAs, as we were interested in studying the complexes as a whole. To check if the interaction of Dicer-2 with its partners was RNA-dependent, we performed IPs of GFP::Dicer-2 with and without RNase A treatment (S4A and S4B Fig). We observed that Dicer-2 interacts with Me31B and eIF4E1 in an RNA-independent manner, as both partners co-immunoprecipitate with Dicer-2 with and without RNase A (Fig 4E). By contrast, the interaction between Rump and GFP::Dicer-2 was partially affected by the RNase A treatment, as we can observe a weaker co-immunoprecipitation of Rump with GFP::Dicer-2^WT and GFP::Dicer-2^Hel1 mutant (S4A Fig). Most strikingly, we observed a loss of the interaction of GFP::Dicer-2 with the protein Syncrip (Syp) upon RNase A treatment (Fig 4E). Of note, a 25KDa band was detected with Syncrip antibody after co-immunoprecipitates with the three GFP::Dicer-2 upon RNase A treatment, but does not correspond to an isoform of Syncrip [52], further investigation is needed to determine if it correspond to a cleavage or degradation product (S4A Fig).

To further investigate the RNA requirement for the interactions of Dicer-2 with candidates lacking corresponding antibodies, we expanded our analysis by performing a second IP-MS experiment with or without RNase A

treatment using adult flies expressing either GFP alone (Ctrl-GFP) or GFP::Dicer-2$^{WT}$ (S4C and S4D Fig). The statistical analyses were performed using the same strategy as described above, in order to compare each condition (GFP::Dicer-2$^{WT}$ with or without RNase A treatment) to its corresponding control (Ctrl-GFP with or without RNase treatment). The results were represented as volcano plots in S4 Fig. As expected, R2D2 was again found to be the top interactant of Dicer-2, with and without RNase A treatment. In addition to R2D2 we can observe the presence of Veneno, Homer, Loqs, FASN2 and PNPase amongst the top interactants of Dicer-2 without RNase treatment (S4C Fig), as in our first analysis (Fig 1). However, although Veneno, Homer and Lost are also amongst the top interactants of Dicer-2 after RNase A treatment, this is not the case for PNPase and FASN2 (S4D Fig). This indicates that, while the potential interaction of Dicer-2 with Veneno, Homer and Lost appears to be RNA-independent, it is probably mediated by RNA for PNPase and FASN2.

Taken together, these results demonstrate that Me31B and eIF4E1 interact with Dicer-2 in an RNA-independent manner, while Rump and Syp do not. Moreover, our MS analysis suggests that Veneno, Homer, and Lost may engage in RNA-independent interactions with Dicer-2. In contrast, PNPase and FASN2 may require RNA to interact with Dicer-2, although these interactions remain to be experimentally validated.

## Identification of new proviral and antiviral factors in drosophila S2 cells

Overall, we established different interaction profiles of Dicer-2 *in vivo*, which may include previously unknown components of the antiviral RNAi pathway or regulators of viral infection. Therefore, we next assessed the impact of these interactants on viral infection by DCV. Our subset of candidates was therefore subjected to an RNAi screen in S2 cells, to test the involvement of the candidates during viral infection. To this end, the expression of the different candidate genes was inhibited in S2 cells using dsRNAs from DRSC Harvard prior to infection by DCV and monitoring of the viral RNA load 20h later (Fig 5A). Whenever possible, two different dsRNAs were used for each candidate gene in order to discount off-target effects, and the dsRNAs targeting RACK1 and AGO2 were used as a proviral and antiviral controls respectively. The data were normalized and analyzed using a linear mixed effect model, and the viral RNA loads of the candidates were compared to that of the dsLacZ control (Fig 5A). Of note, knockdown of AGO2, using dsRNA, does not lead to a significant antiviral response, whereas the endo-siRNA pathway is impacted successfully, likely due to viral charge and time points use in our experiment setting (Figs 5B and S5A). However, we found that some of the candidates, eIF4G1 (2/2 dsRNAs) and Rin (1/2 dsRNA), exhibited a significant antiviral phenotype against DCV infection. The viral RNA increase in KD of eIF4G1 is correlated with accumulation of DCV RdRp protein (S5B and S5C Fig). Notably, both proteins physically interact with Dicer-2 (Fig 4C and 4D), underscoring their potential role in the antiviral RNAi pathway. Moreover, knockdown of *lig* led to a significantly decreased DCV RNA load suggesting a proviral function of the Lig protein (2/2 dsRNA). Moreover, we observed that *rump* knockdown (KD) led to one of the highest viral loads amongst the candidates tested, although it did not reach statistical significance in our assay conditions (Fig 5A). Rump is a hnRNP M homologue that binds to *nos* and *osk* mRNAs and has hitherto not been implicated in antiviral immunity. The other proteins for which the interaction with Dicer-2 was validated (Me31B, eIF4E1 and Syp) do not seem to have any impact on DCV RNA load at all, suggesting that they might be involved in the other functions of Dicer-2. Interestingly, co-transfected GFP-Me31B and RFP-Dicer-2 appeared to co-localize in punctuated cytoplasmic granules in S2 during DCV infection (S5D Fig). To assess the impact of the main candidates (eIF4G1, Rin, Rump, Homer, Rm62, Me31B and AGO2) on the endo-siRNA pathway, we then used an endo-siRNA sensor assay, wherein the *Renilla* and firefly luciferases were used as a reporter for endo-siRNA2 activity and a transfection control, respectively (Fig 5B). However, all candidates except AGO2 failed to inhibit endo-siRNA2 activity, which would suggest that the candidates are not necessary for the endogenous siRNA-dependent pathway. Overall, a high proportion of the candidate genes show an increase of DCV viral load after KD although not statistically significant, suggesting that they could still play a minor role in antiviral immunity in drosophila S2 cells.

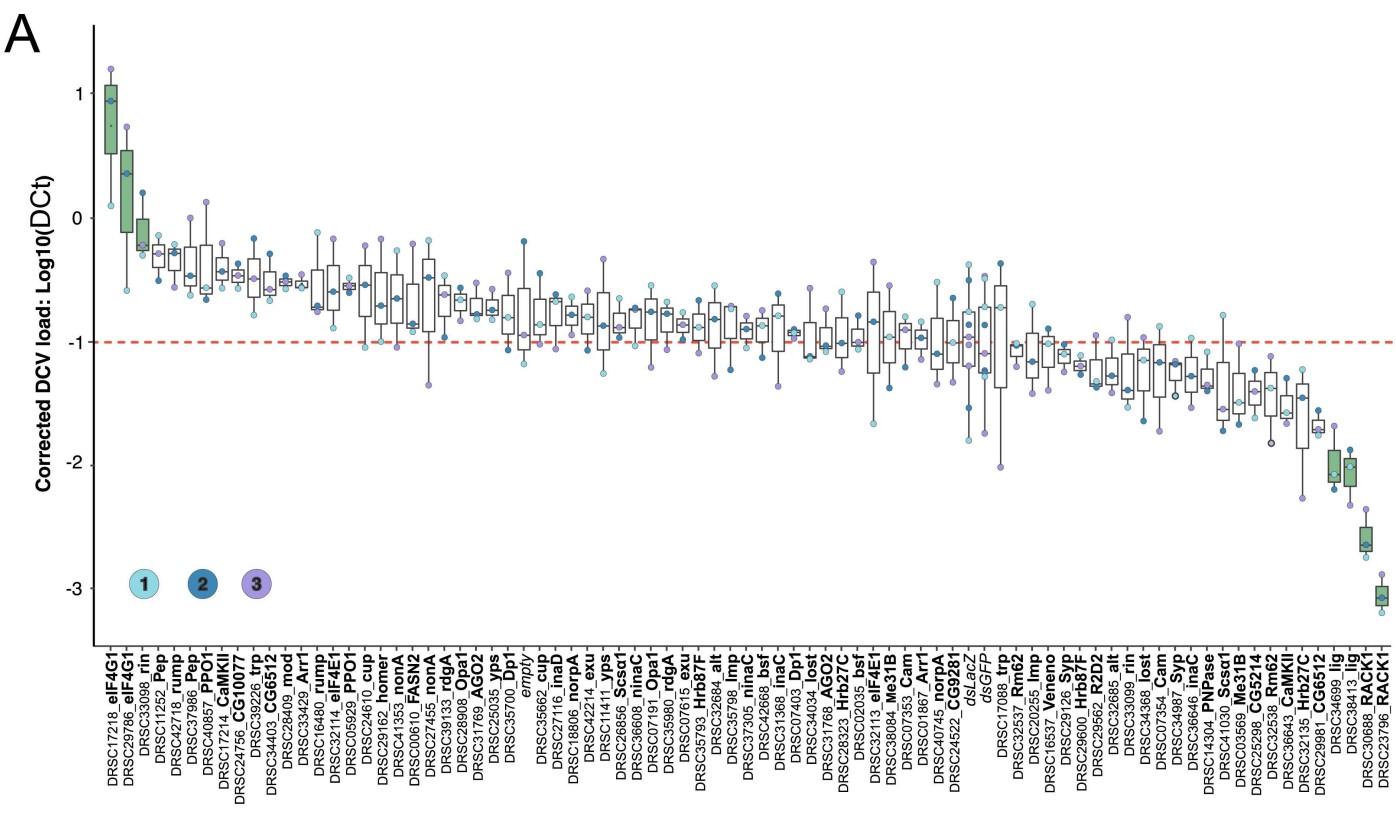

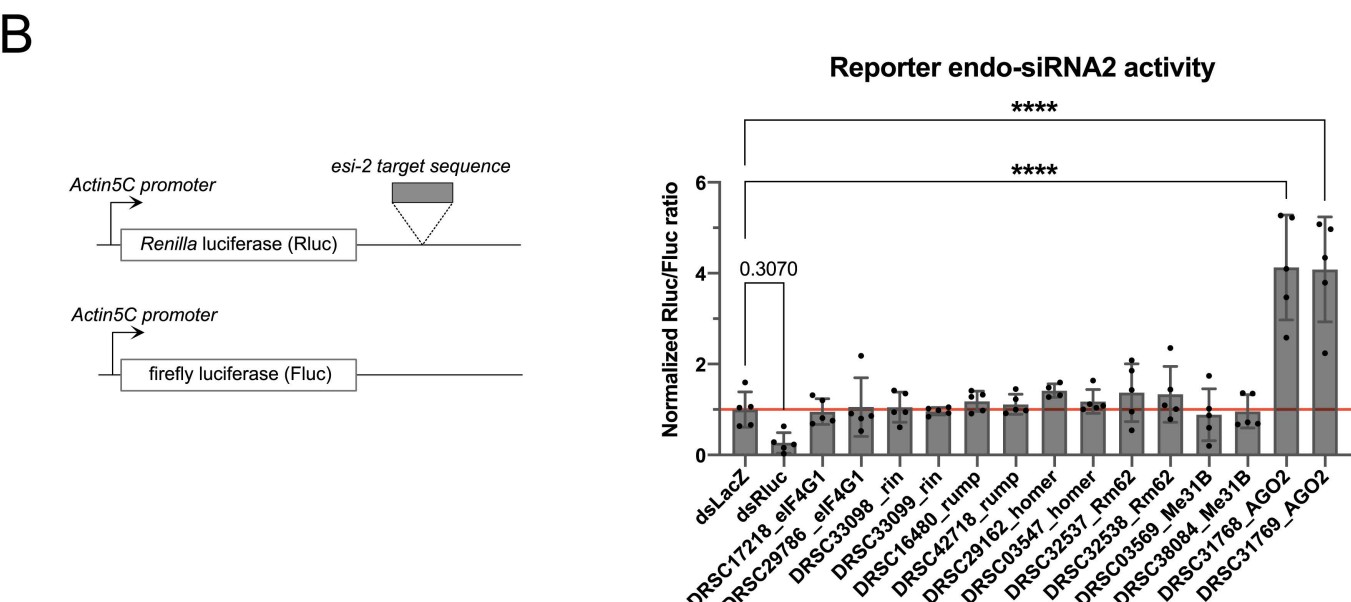

**Fig 5. Role of the candidates in antiviral immunity in drosophila S2 cells.** (A) Knockdowns (KD) of gene expression were performed by soaking drosophila S2 cells with dsRNA targeting the candidate genes during 3 days. On day 3, cells were infected with DCV at 0.001 MOI for 20h. Box plot representing the DCV viral RNA load relative to housekeeping gene RP49 (ΔCt) of S2 cells after KD of the top 15% candidates followed by DCV infection. This ratio was corrected using a Linear Mixed effect model to account for variation between plate and column position. The boxes represent interquartile range (IQR), the inner line represents the median, and the whiskers extend to the minimum and maximum values. The dsRNAs targeting candidate genes are represented with the DRSC number and the gene name. Negative controls are infected S2 cells that were not treated with any

dsRNA (empty), dsRNA against GFP, dsRNA against LacZ. In addition, dsRACK1 was used as a proviral control. The three independent experiments are indicated by the color of the dots. Statistical comparisons between treatment groups and the control group (dsLacZ) were performed using *t*-tests with *p*-values adjusted by Dunnett's method to account for multiple comparisons. The significant antiviral and proviral candidates are in green with an adjusted *p*-value<0.05. (B) Schematic representation of endo-siRNA2 reporter constructs. *Renilla* luciferase is flanked in the 3'UTR by the esi-2 target sequence. Firefly luciferase is used as transfection control. Both constructs are under of the constitutive *Actin5C* promoter. Candidates depleted S2 cells were transfected with endo-siRNA2 *Renilla* luciferase reporters together with firefly luciferase (transfection control). The ratio *Renilla* luciferase counts (Rluc) with firefly luciferase counts (Fluc) were normalized to dsLacZ treated S2 cells (n=5 independent experiments). Mean±standard deviation (SD). Statistical significance was analysed using one-way ANOVA followed by Dunnett's multiple comparison test. "****" for *p*<0.0001.

## Identification of new proviral and antiviral factors *in vivo*

We next evaluated the impact of a smaller subset of the interactants *in vivo*, corresponding to the top 10% of the SAINT analysis but excluding candidates for which the knockdown (KD) was lethal *in vivo* (e.g. eIF4G1), on DCV infection in the context of the whole organisms. A knockdown of the different candidate genes was induced through temperature shift in adult flies using the [*actin*-Gal4; *tub*-Gal80^TS] system, and the flies were then injected with DCV (Fig 6A). Survival was assessed by counting flies every day for 20 days and computing the hazard ratios for each KD fly line, which corresponds to the chances of the different KD flies dying compared to the control shmCherry flies (Figs 6B and S6). In addition, viral RNA load was also measured by RT-qPCR at 1 and 2 dpi after injection (Fig 6C).

Unfortunately, the strongest candidate highlighted by the in cells screen, eIF4G1, could not be tested *in vivo* as its knockdown is lethal in the whole fly. The most striking impact on viral infection *in vivo* was observed for Rin. Indeed, although the hazard ratio for Rin in mock-infected flies is close to 1, meaning that the injection of TRIS alone does not impact survival in Rin KD flies, the hazard ratio for DCV-infected Rin KD flies is significantly higher than 1, meaning that Rin KD has a negative impact on survival after DCV infection (Fig 6B). This correlates with the higher DCV RNA load observed *in vivo* (Fig 6C), and in cells (Fig 5A). Taken together, these data strongly suggest that Rin is involved in the antiviral response against DCV in *Drosophila melanogaster*.

Similarly, the hazard ratio for Rump KD flies is higher than 1 after infection with DCV but not TRIS (Fig 6B), and is correlated with a higher DCV RNA load *in vivo* (Fig 6C), confirming the trend observed in cells (Fig 5A). Thus, Rump may also be involved in the antiviral response against DCV. We also tested the role of Lig, however *in vivo* data monitoring survival (Fig 6B) or viral RNA load (Fig 6C), did not confirm the proviral effect observed in cell culture (Fig 5A). Amongst the other candidates tested, we observed a strong proviral impact of another candidate, Rm62, on DCV RNA load (Fig 6C). However, Rm62 KD also highly impacts survival in non-infected conditions, which suggests that Rm62 impacts the general stress response after injection.

## Discussion

The crucial role of a few known protein partners of Dicer-2 in regulating or facilitating its activity has been established [14,32,64]. Yet, we are far from truly understanding the many functions of this multifunctional protein. In order to establish the protein network of Dicer-2 *in vivo* in different conditions, we have used different tagged GFP::Dicer-2 constructs and an IP-MS approach. We have confirmed the reliability of this approach by the identification of the usual key players of RNAi, also found in other interactomes of RNAi proteins [28,65–68], e.g., R2D2, AGO1, AGO2 and Loqs. Moreover, we were also able to observe recurring interactions compared to the networks established in other studies. Indeed, we were able to confirm the interaction of Me31B, also known as DDX6, with Dicer-2 in drosophila *in vivo*, and this protein was also identified in mammalian cells as a potential interactant of AGO2 [65]. The protein Larp4B, which was found in the list of 27 proteins interacting with all three GFP::Dicer-2 lines independently of the conditions, provides another example. This nucleic acid binding protein interacts with Dicer-2 in the embryo and has been shown to be antiviral against DCV, CrPV,

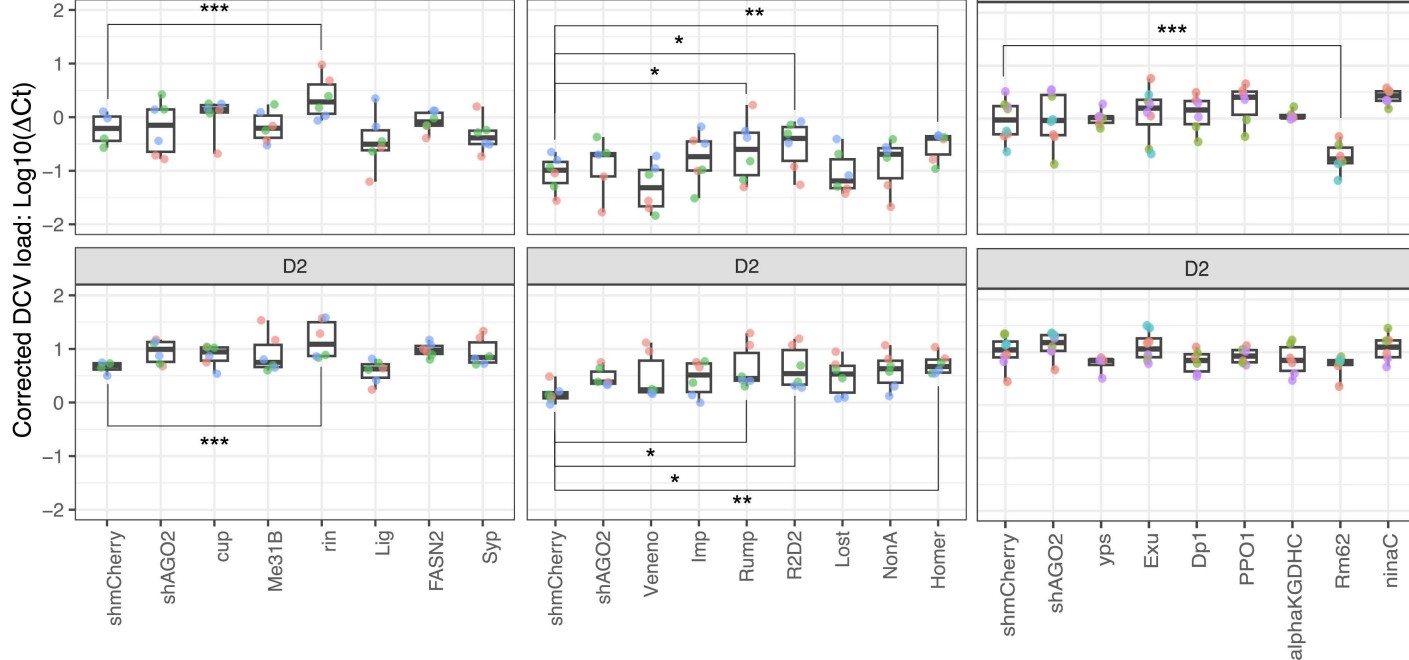

**Fig 6. Role of the candidates in antiviral immunity in adult drosophila flies.** (A) Schematic representation of the experiment procedure. Knock-down (KD) of gene expression were established by crossing inducible UAS-dsRNA or UAS-shRNA for each candidate gene with an ubiquitous Gal4 driver (actin5C-Gal4) under the control of thermosensitive ubiquitous Gal80 (tub-Gal80$^{TS}$) during five days at 29∞C. Adult flies were injected with DCV 50pfu and analysed for survival or DCV 500pfu followed by RT-qPCR at days 1 and 2 post-infection. Data were collected from three or four independent experiments, each comprising three groups of 20 flies. (B) Hazard ratios of the different UAS-dsRNA/shRNA lines after mock-infection or DCV-infection.

(C) Box plot representing the DCV viral RNA load relative to housekeeping gene RP49 (ΔCt) at day 1 and day 2 post-infection with DCV 500 pfu. This ratio was corrected using a Linear Mixed effect model to account for variation between experiments. The boxes represent interquartile range (IQR), the inner line represents the median, and the whiskers extend to the minimum and maximum values. Triplicate experiments represented in three batches. Statistical comparisons between treatment groups and the control group (shmCherry) were performed using $t$-tests with $p$-values adjusted by Dunnett's method to account for multiple comparisons. Significant viral load compared to shmCherry line are represented as follows: "***" for adj.$p$-val < 0.001, "**" for adj.$p$-val < 0.01, "*" for adj.$p$-val < 0.05.

SINV and VSV [69,70]. Moreover, we have been able to identify proteins that were shown to interact with Dicer in cells of other species, e.g., DHX9/Mle and proteins from the Heat shock protein family 70 [28,68].

By contrast with the previous interactomes of Dicer-2 in other species this is, to our knowledge, the first interactome of Dicer-2 performed *in vivo* in adult flies. Moreover, the study of the impact of two point mutations affecting key domains of the protein has allowed us to highlight proteins interacting with only one, two, or all GFP::Dicer-2 samples. Our results reveal a list of 27 proteins interacting with all three GFP::Dicer-2 lines in non-infected and DCV-infected conditions, which indicates a very stable interaction, not disturbed by those specific mutations, and reproducible as it was observed with three different GFP::Dicer-2 fly lines. In addition, we also identified proteins interacting only with the WT version of GFP::Dicer-2. These proteins may correspond to factors of a functional Dicer-2 complex, actively producing siRNAs. On the contrary, we identified proteins interacting only with the two mutants, which sequestrate the dsRNA in an unproductive complex [19]. These proteins may correspond to factors interacting with Dicer-2 in a transient manner. For example, we have noticed a stronger interaction of the two GFP::Dicer-2 mutants with a transposable element called Roo/ORF, which is not present for the WT version of Dicer-2. Keeping in mind that transposable elements participate in the reverse-transcription of viral RNA and have been proposed to play a role in systemic antiviral RNAi [71], the study of Roo/ORF could prove interesting. We also provide a list of proteins whose interaction with Dicer-2 depends on the presence of a functional helicase or RNase III domain.

Beyond the siRNA pathway, Dicer-2 plays a crucial role in gene regulation, controlling the expression of specific mRNA subsets, stress resistance, and life span [72–74]. Accordingly, some of the Dicer-2 partners we highlighted are proteins enriched during oogenesis, including three of the proteins for which the interaction with Dicer-2 was confirmed by WB: Me31B, eIF4E1 and Rump. This suggests that Dicer-2 may regulate specific mRNA subsets in an RNAi-independent manner during this process [72,73]. By looking deeper into those candidates, we noticed that most of them (e.g., Dp1, Lost, Cup, eIF4E1, Me31B, Imp, Syp, Bsf, Rump, Exu & Yps) were associated with mRNA localization regulation mechanisms, like the regulation of *nanos* or *oskar* mRNA translation. Indeed, Me31B has been shown to form a complex with Tral, eIF4E1, Cup and PABP involved in RNP-mediated translational repression of maternal mRNAs during oogenesis and embryogenesis [60]. It has also been shown to form a complex with Exu and Yps involved in the translational silencing of mRNAs (such as *oskar*) during their transport to the oocyte [75]. The presence of eIF4E1, Cup, Exu and Yps in the candidates from the SAINT analysis could suggest that Dicer-2 could interact with those two complexes formed by Me31B. An impact of Dicer-2 in oogenesis could explain the decreased fertility of *dicer-2* null mutants. Recently, Dicer-2 has been shown to form a complex with Ataxin-2/Tyf forming a noncanonical cytoplasmic polyadenylation complex [69]. In addition, Rump and Lost, which interact together, have also been shown to be involved in mRNA regulation in the oocyte [76,77] and this is also the case for Syp [52]. This would not be the first time that an RNAi-independent role in mRNA regulation of Dicer-2 is reported, as previous studies have already reported an RNAi-independent role of Dicer-2 in the regulation of mRNA expression, and more specifically in the activation of the expression of *Toll* and *R2D2* mRNAs through cytoplasmic polyadenylation [72,73]. Interestingly, another small RNA factor, Aubergine (Aub), has been implicated in both mRNA localization regulation and Wispy-related mRNA regulation mechanisms [78] and has been implicated as a *nanos* mRNA localization factor in a mechanism implicating Rump [79]. If Dicer-2 does have a link with the regulation of mRNA

expression during oogenesis, it would underline the interconnection between small RNA factors and RNAi-independent mRNA regulation mechanisms. Furthermore, while our analysis highlights the connection between Dicer-2 and oogenesis, which we were able to detect even though it was based on a mixed population of male and female tissues, it might not capture other tissue-specific interactions. Investigating specific tissues, such as the fat body or gut, which are targets of viral infection, could potentially uncover differential interactions and functional roles of Dicer-2.

An important objective of this study was to analyze the impact of DCV infection on the protein network of Dicer-2, and the contribution of the interactants identified to the control of the infection. These experiments have highlighted four candidates (Rump, Lig, Rin and Rm62).

Rm62 is a member of the DDX5/Dbp2 subfamily of DEAD-box helicases [80], which have been shown to be important in virus-host interactions [1]. In particular, some DEAD-box helicases play a role in RNAi and bind dsRNA in an ATP-dependent manner [81,82]. Rm62 recognizes stem loop structures and was previously shown to facilitate miRNA processing and antiviral defense against the Rift Valley fever virus (RVFV) in drosophila both in cells and *in vivo* [83]. Therefore, our results could suggest that Rm62 may be recruited to a stem loop structure of the DCV RNA to facilitate its replication.

The protein Rin belongs to the GTPase activating protein (SH3 domain) binding protein family (also known as G3BPs) a family of RBPs that regulate gene expression in response to environmental stresses. In unstressed cells, Rin plays a role in the stabilization of target mRNAs and upregulation of their translation [84]. This regulation can happen inside or outside stress granules (SGs), of which Rin is a core component, or outside of them [85]. As SGs are induced in response to viral infections, some viruses sequestrate G3BPs [86] while others such as the poliovirus (*Picornaviridae* family) target and cleave them in order to disrupt the formation of stress granules [87]. This is interesting as DCV is a picorna-like virus. In some cases, e.g., in alphaviruses, a conserved interaction with the viral proteins Nsp3 and SG proteins [88] not only prevents SG formation but also act in favor of the virus, as in the case of the Rin-Nsp3 interaction for the chikungunya (CHIKV) virus [89]. CHIKV is able to utilize Rin in order to increase infection rate and transmissibility. Interestingly, although we saw an antiviral effect of Rin and a proviral effect of Lig during DCV infection, the two proteins have been shown to act in concert to regulate cell proliferation during development [90]. In addition, the human homologue of Lig, UBAP2L, was identified after pull-down of the human homologue of Rin, G3BP1, in HEK293T cells [91]. A possible explanation to these opposite effects could be that the disruption of this interaction in Lig KD flies allows Rin to focus on its antiviral function, and the absence of Lig would therefore be detrimental to the virus.

Moreover, viral infection triggers an inducible antiviral response, with Dicer-2 playing a dual role by activating the siRNA pathway and inducing gene expression [11,92]. While the siRNA pathway has long been recognized as a key antiviral defense mechanism in invertebrates, recent studies revealed that dsRNA sensing can also activate additional immune pathways, particularly the cGLR/STING pathway [93,94]. In the absence of infection, RNAi components are ubiquitous and mediate a cell-autonomous response [95], whereas cGLR/STING activation leads to a systemic response [96,97]. DCV infects a wide range of tissues in adult flies, yet STING::GFP induction is predominantly detected in non-infected cells, indicating a signal transmitted from the infected cells to naive cells [97]. Further research should focus on unraveling the molecular interactions between RNAi and cGLR/STING pathways in antiviral defense.

In conclusion, we have been able to establish distinct interaction profiles of Dicer-2^WT and its mutants *in vivo*, which were strongly altered by the DCV-infection. By doing so, we have established several categories of candidate proteins, differentially enriched across the different experimental conditions. We were then able to confirm the interaction of Dicer-2 with six of the candidates: three *in vivo* (eIF4E1, Rump and Syp), two in S2 cells (Rin and eIF4G1), and one both *in vivo* and in S2 cells (Me31B). Finally, we could highlight the anti- or proviral phenotype of five candidates, in S2 cells (eIF4G1, Rin and Lig) and/or *in vivo* (Rin, Rump and Rm62). Overall, this work has produced a resource and a large amount of data that is now available for the community. Our network hints at a potential involvement of Dicer-2 and several candidates in different mechanisms, in both mRNA regulation and antiviral immunity.

## Materials and methods

### Flies, drosophila S2 cells and virus

The GFP::Dicer-2 fusion drosophila fly lines were obtained by crossing [*w*$^{IR}$; *dicer-2*$^{L811fsx}$/*CyO*] virgins with: [*w*$^{IR}$; *dicer-2*$^{L811fsx}$/*CyO*; ubi > GFP::Dicer-2$^{WT}$] males (GFP::Dicer-2$^{WT}$), [GFP::Dicer-2$^{Hel1}$: *w*$^{IR}$; *dicer-2*$^{L811fsx}$/*CyO*; ubi > GFP::Dicer-2$^{G31R}$] males (GFP::Dicer-2$^{Hel1}$), or [*w*$^{IR}$; *dicer-2*$^{L811fsx}$/*CyO*; ubi > GFP::Dicer-2$^{E1371K/E1617K}$] males (GFP::Dicer-2$^{RNaseIII}$). The first two fly lines (GFP::Dicer-2$^{WT}$ and GFP::Dicer-2$^{Hel1}$), were established as described previously [98]. [*w*$^{IR}$; *dicer-2*$^{L811fsx}$/ *dicer-2*$^{L811fsx}$; ubi > GFP::Dicer-2$^{WT}$] phenocopy CantonS flies for RNAi pathway [46]. GFP::Dicer-2$^{RNaseIII}$ line was established in the same manner with the following mutations in the Dicer-2 sequence: E1371K and E1617K, thus inactivating the activity of its RNase III domain. The control fly lines used were CantonS flies and GFP expressing flies, obtained by crossing *P{UAS-GFP.S65T}Myo31DF[T2]* line (BDSC #1521) under the control of the *actin5C* promoter (BDSC #25374). All flies used were Wolbachia free. All experiments were performed with an equivalent number of males and females.

S2 cells were grown in Schneider's medium (Biowest) supplemented with 10% fetal calf serum, Glutamax (Invitrogen) and Penicillin/Streptomycin (100x mix, 10 mg/mL/ 10000 U, Invitrogen).

DCV virus stock was produced as described [6]. Detailed protocols and additional experimental procedures were provided in the S1 Text.

### Viral infection and immunoprecipitations for MS analysis

Infections were performed on 3- to 5-day-old flies. Forty flies (20 females and 20 males) of each phenotype were injected with TRIS (10 mM, pH 7.5) or DCV (500 PFU) with 4.6 nL of TRIS (10 mM pH 7.5) or DCV (500 PFU) by intrathoracic injection (Nanoject II apparatus; Drummond Scientific). Flies were kept at 25°C for either 2 or 3 days, and then collected in Precellys tubes with zirconium beads and frozen overnight at -80°C. The flies were first shredded at 10°C without any buffer and then a second time with 1 mL lysis buffer (30 mM HEPES KOH pH 7.5, 50 mM NaCl, 2 mM Mg(OAc)2, 1% NP40, 2X cOmplete Protease Inhibitor Cocktail [Roche]). If the impact of RNase A on the interactants was tested, samples were separated in two halves and one half was treated with 15 µg RNase A for 30 min at 4°C. Samples were then centrifuged and the rest of the protocol was performed in the same manner as for the other MS experiment. After centrifugation, 20 µL (0.05%) of the protein supernatant was kept to be loaded on a gel later on (S1A Fig) and the rest was incubated with anti-GFP bead (Miltenyi) for 40 min at 4°C on a spinning wheel. The protein samples were then added onto a microcolumn placed on an uMACS separator (Miltenyi) after equilibration of the microcolumn with lysis buffer. Three washes were performed using wash buffer (30 mM HEPES KOH pH 7.5, 50 mM NaCl, 2 mM Mg(OAc)2, 0.1% NP40, 2X cOmplete Protease Inhibitor Cocktail [Roche]) and the elution was performed using elution buffer (Miltenyi) heated to 95°C. Five microliters (25%) of the elution was kept to confirm the IP on a WB and the rest was immediately used for mass spectrometry analysis. The MS experiments have been conducted in triplicate.

### Mass spectrometry analysis

Proteins were digested with sequencing-grade trypsin (Promega, Fitchburg, MA, USA). Each sample was analyzed by nanoLC-MS/MS on a QExactive+ mass spectrometer coupled to an EASY-nanoLC-1000 (Thermo-Fisher Scientific, USA) as described previously [99]. Data were searched against the Drosophila melanogaster UniprotKB sub-database with a decoy strategy (UniprotKB release 2018_02 and 2022_01, taxon 7227, 42551 forward protein sequences). The correlation between spectrum matches for common baits of the two different MS datasets was assessed, as the two MS experiments were performed using different UniprotKB releases, and was greater than 0.999. Peptides and proteins were identified with Mascot algorithm (version 2.5.1, Matrix Science, London, UK) and data were further imported into Proline v1.4 software (http://proline.profiproteomics.fr/). Proteins were validated on Mascot pretty rank equal to 1, and 1% FDR on both

peptide spectrum matches (PSM score) and protein sets (Protein Set score). The total number of MS/MS fragmentation spectra was used to quantify each protein from at least three independent biological replicates. The mass spectrometric data were deposited to the ProteomeXchange Consortium *via* the PRIDE partner repository with the dataset identifier PXD038898 and 10.6019/PXD038898. Raw data and Script are available (https://doi.org/10.6084/m9.figshare.27909918).

## Statistical post-processing of the MS data and bioinformatics analysis

For each fly line, statistical post-processing of the data was performed through R (v4.2.1), using the IPinquiry4 package [100]. After a column-wise normalization of the data matrix, the spectral count values were submitted to a negative-binomial test using an edgeR GLM regression as well as the msmsTests R package (release 3.20, [101]). For each identified protein, an adjusted *p*-value (adjp) corrected by Benjamini–Hochberg was calculated, as well as a protein fold-change (FC), representing enrichment compared to the controls (Ctrl-WT + Ctrl-GFP in the case of Figs 1 and 2, and Ctrl-GFP in the case of S4 Fig). The results are presented in a Volcano plot using protein log2 fold changes and their corresponding adjusted log10 *p*-values to highlight upregulated proteins. Duplicate genes were manually removed and the results of the lists of genes enriched in the different conditions were put in JVenn to produce Venn diagrams [102], or in R using the Pheatmap function to produce heatmaps representing raw number of spectra.

An overall analysis of the data was performed using the Significance Analysis of INTeractome (SAINTexpress, v3.6.1) tool for interaction scoring. The data was then represented using Prohits-viz (primary filter 0.01, secondary filter 0.05, Ward clustering method and Camberra distance metric). A protein network was represented using the STRING tool in Cytoscape (version v3.9.1), after inputting the top 15% of the SAINT analysis candidates.

## Confirmation of the interactions by immunoblot analysis

The input and elution samples were obtained after immunoprecipitation using anti-GFP beads (Miltenyi or agarose beads from Chromotek). If necessary, injections followed by RNase A treatment or not were performed as described above. Protein extracts were separated on a Biorad gel and transferred to a nitrocellulose membrane. Membranes were then blocked in TBST containing 5% milk powder for 1h at RT and incubated overnight at 4°C with the different primary antibodies listed in S3 Table. After washing, the corresponding secondary antibodies fused to horseradish peroxidase (HRP) were added to the membrane for 2h at RT. Membranes were then washed and visualized with enhanced chemiluminescence reagent (GE Healthcare) in a ChemiDoc (Bio-Rad) apparatus. Detailed protocols and additional experimental procedures are provided in the S1 Text.

## RNAi screen in S2 cells

After clustering of the 288 candidates highlighted by SAINT analysis after LC-MS/MS, the top 15% were selected to be part of a RNAi mini-screen. The dsRNAs were ordered at Drosophila RNAi Screening Center (DRSC) at Harvard Medical School. 47 dsRNAs were ordered in total, amongst which *dsAGO2* and *dsRACK1*, which were used as antiviral and proviral controls, respectively. Moreover, two negative controls (*dsGFP* and *dsLacZ*) and an RNAi knockdown (KD) efficiency control (*dsThread*) were placed in triplicate for each plate and put at different positions on the plate. *dsGFP* and *dsLacZ* were used to detect and normalize column and/or line effect, and *dsThread* controls were used to check that RNAi was working properly for each plate. Cells were seeded in 96-well plates and incubated with 1μg of dsRNA in FBS-free Schneider medium for 4h. After this starving period, normal S2 medium containing FBS was added, and the cells were incubated for 3 days before infection. Infections were performed at 25°C with MOI of 0.01 for 20h, after adsorption on ice for 1h. Cell lysis and reverse transcription were then performed using the Cells-to-CT kit (ThermoFisher Scientific) according to the manufacturer protocol, and used to perform quantitative real-time qPCR using the iTaq Universal SYBR Green Supermix (Biorad). The primers used are listed in the S3 Table. After calculating the $2^{\Delta Ct}$ for each sample, different mixed

effect linear models were tested to search for bias in the data (lme4 R package (version 1.1-29)). As variations in plate, rows and columns had a significant impact after ANOVA, they were used as random factors in the final mixed effect model chosen. These linear mixed models can be represented as follows:

$$y = X\beta + Z\omega + \varepsilon$$

Where y contains the measured values (in our case, ΔCt values), X is the fixed effect component of these values (in our case, the impact of the gene tested by KD on the ΔCt observed), Z is the random effect (in our case, the plate and column variability) and ε is the residuals (everything that cannot be explained by the fixed and/or random effects: in our case, bio-logical variability). In order to normalize our data while still retaining the biological variability of the experiments, we plotted Xβ + ε, which represent the observed values minus the random effects caused by the factors identified with our models. Statistical significance then was calculated by comparing the corrected ratios of each dsRNA to that of the dsLacZ control using the emmeans function. Raw data and Script are available (https://doi.org/10.6084/m9.figshare.27909918).

### Genetic UAS-RNAi screen in an *actin5C*-Gal4;*tub*-Gal80^TS system

The top 15% candidates highlighted by the SAINT analysis were selected for the genetic screen *in vivo*. KK and GD inverted repeat transgenic fly lines for each candidate gene were acquired from the VDRC stock center (S4 Table), and *shmCherry* (BDSC #35787) and *shAGO2* (BDSC #34799) were used as controls, in addition to the respective KK and GD control lines. Transgenic males containing the inverted repeat of the target gene under the control of Gal4 regulated upstream activating sequence (UAS) were crossed with virgin females [*actin5*C-Gal4/*CyO*; *tubulin*-Gal80^TS] at 18°C. The F1 generation (2 times 5 males and 5 females) was placed at 29°C for 5 days to induce the knockdown of candidate genes. All experiments were subsequently performed at 29°C. Infections were then performed as described above. The flies were then either counted every day to assess survival or collected at 1 or 2 dpi for RT-qPCR. In this case, three males and three females per condition were collected. Total RNA from the flies was then extracted using Trizol-chloroform, and 500 ng of total RNA was reverse transcribed using the iScript gDNA Clear cDNA Synthesis Kit (Biorad) according to the manufacturer's instructions, and used to perform quantitative real-time qPCR using the iTaq Universal SYBR Green Supermix (Biorad), on a CFX384 Touch Real-Time PCR platform (Bio-Rad). The qPCR primers used are listed in S2 Table.

### Normalization and qPCR/survival analysis

All statistical analyses were performed in R (version 3.6.1). To perform the qPCR analysis, after calculating the $2^{\Delta Ct}$ for each sample, different mixed effect linear models were tested to search for bias in the data (lme2 R package (version 1.1-29)). Each batch was analyzed independently, and the data was tested for normality and homoscedasticity. As variations in replicates had a significant impact after ANOVA, this parameter was used as a random factor in the final mixed effect model chosen. Again, these linear mixed models can be represented as follows:

$$y = X\beta + Z\omega + \varepsilon$$

Where y contains the measured values (in our case, ΔCt values), X is the fixed effect component of these values (in our case, the impact of gene tested using transgenic fly lines and the days post infection, or "dpi", on the ΔCt observed), Z is the random effect (in our case, the variability between experiments) and ε is the residuals (everything that cannot be explained by the fixed and/or random effects: in our case, biological variability). In order to normalize our data while still retaining the biological variability of the experiments, we plotted Xβ + ε, which represent the observed values minus the random effects caused by the factors identified with our models. Moreover, dependence between the "Gene" and "dpi"

fixed factors was tested and found to be true for Batch 3; the models were adjusted accordingly. Statistical significance was calculated by comparing the corrected ratios of each fly line to that of the shmCherry control using the emmeans function. P-values were adjusted using the Dunnett method. For survival analysis, the hazard ratio for each candidate was computed using a Cox Proportional-Hazards Model. In addition, Kaplan-Meier curves for each gene were produced using the Survival package, and the corresponding logrank tests were performed to compare each candidate to shmCherry. Raw data and Script are available (https://doi.org/10.6084/m9.figshare.27909918).

## Supporting information

**S1 Fig. Fly lines used for the establishment of the Dicer-2 interactome *in vivo*.** (A) Immunoblot showing the presence of the Dicer-2 in all the samples before immunoprecipitation (IP) using anti-GFP beads, and in the GFP::Dicer-2 samples after IP. The band for the GFP::Dicer-2 fusion protein is slightly higher than Dicer-2 due to the added size molecular weight of the GFP, as expected. Actin, which does not interact with Dicer-2, can be seen in the input but not in the elution. (B) The color of the eyes of $w^{IR}$ flies allow the monitoring of RNAi efficiency. The *dicer-2^null^* and *dicer-2^rescue^* flies are described in Kemp *et al.*, 2013 [6]. Flies without the transgene normally have red eyes (see Ctrl-WT and Ctrl-GFP). When the flies contain the $w^{IR}$ transgene, if RNAi is efficient, it induces KD of the *white* gene and results in a white eye phenotype (see *dicer-2^rescue^*) and if RNAi is inefficient the KD is not effective and eyes are red like WT flies (see *dicer-2^null^*). RNAi works normally in GFP::Dicer-2WT, but not GFP::Dicer-2Hel1 or GFP::Dicer-2RNaseIII flies. Photos, by C. Meignin, are under CC BY 4.0 license. (C) Multidimensional Scaling (MDS) analysis showing the five genotypes in non-infected (NI) and infected (DCV) conditions tested in triplicate. This plot visualizes the overall distances between samples with normalization based on the total count number. Control lines (Crtl-WT and Crtl-GFP) cluster together but distinctly from the Dicer-2::GFP samples, reflecting their unique characteristics. Non-infected and DCV infection show no clear separation. See S1 Text.
(TIFF)

**S2 Fig. Impact of the Dicer-2 mutations on the RNP network of Dicer-2.** Heatmap representing the number of spectra of the proteins enriched in any of the three GFP::Dicer-2 lines (Fig 1C-E), separated by the categories of the Venn diagram in Fig 1F.
(TIFF)

**S3 Fig. Impact of the the DCV infection on the RNP network of Dicer-2.** (A) Venn diagram showing the number of proteins identified in all three GFP::Dicer-2 lines in mock-condition (Fig 1F, 52 proteins) and/or all proteins identified in all GFP::Dicer-2 lines in DCV-infected adult flies (Fig 2D, 83 proteins). (B-D) Heatmaps representing the number of spectra of the proteins enriched in the different categories depicted in (A). Clustering was performed using the Ward method.
(TIFF)

**S4 Fig. The RNA-dependent protein network of Dicer-2.** (A) Immunoprecipitation of (A): GFP::Dicer-2^WT^ or, GFP::Dicer-2^Hel1^ or GFP::Dicer-2^RNAseIII^, in comparison to wild-type CantonS flies (Ctrl-WT) with (+) or without (-) RNase A treatment. Representative of three independent experiments (n = 3) except for Dicer-2^RNaseIII^ (n = 1). (B) Urea/Acrylamide gel showing total RNA after no treatment (-) or treatment with 15 μg of RNase A (+). Adult flies were injected or not with DCV and protein extraction was performed in the same manner as for the IPs. Instead of proceeding with the IP, total RNA was extracted after no treatment or treatment with RNase A to confirm the efficiency of the RNase treatment. (C, D) Volcano plots representing the fold changes and adjusted *p*-value of the GFP::Dicer-2^WT^ partners *versus* the control line (Ctrl-GFP) either without (C) or with (D) RNase A treatment. Fold-changes and adjusted *p*-values (adjp) were obtained using a negative binomial test and *p*-values were corrected by the Benjamini-Hochberg method to obtain adjusted *p*-values. All the proteins with fold change > 2 and an adjusted *p*-value < 0.01 are represented in blue.
(TIFF)

**S5 Fig. Impact of the Dicer-2 partners on endo-siRNA pathway and during DCV infection.** (A) Candidates depleted S2 cells were transfected with endo-siRNA1 *Renilla* luciferase reporters together with firefly luciferase (transfection control). *Renilla* luciferase counts (Rluc) were normalized using firefly luciferase counts (Fluc) (n = 3 independent experiments). Mean ± standard deviation (SD). Statistical significance was analysed using one-way ANOVA followed by Dunnett's multiple comparison test. "****" for $p < 0.0001$. (B) Immunoblot of DCV RdRp in S2 cells after knockdown of Dicer-2's partners. DCV RdRp is only detected in infected cells excepted dsRNA RACK1 treated cells which served as positive control. Representative of two independent experiments. (C) S2 cells treated with dsRNA targeting eIF4G1 inhibits the expression of HA-eIF4G1. Representative of two independent experiments. Actin antibody is used as loading control. (D) Co-expression of RFP-Dicer-2 and GFP-Me31B reveals co-localisation in S2 cells during DCV infection. Arrows indicate cytoplasmic granules with the two proteins. dsRNA are stained with J2 antibody. Transfected cells were infected at DCV MOI1 during 24h. blue = DAPI. Representative of two independent experiments. scale = 20μm.
(TIFF)

**S6 Fig. Role of the candidates in antiviral immunity in adult drosophila flies.** Survival analysis of Rin, Rump, Lig and Rm62 knockdowns *in vivo* after infection with DCV 50 pfu. Data were collected from three or four independent experiments, each comprising three groups of 20 flies (n = number of individual flies is indicated). Log-rank test with Benjamini-Hochberg (BH) correction was used for multiple comparisons of survival curves in each batch. Non infected (NI) and DCV infected conditions for each survival were compared to shmCherry line. The *p*-values are represented as follows: "***" for adj.*p*-val < 0.001, "****" for adj.*p*-val < 0.0001 and "ns" for non significant.
(TIFF)

**S1 Table. Genetic information and raw mass spectrometry data.**
(XLSX)

**S2 Table. Candidate list from SAINT analysis.** List of 288 proteins that were highlighted using the SAINTexpress tool. The baits are represented in green, two viral proteins are in orange, the top 10% candidates are in blue, and the top 15% are in purple.
(TIFF)

**S3 Table. Antibodies used.**
(TIFF)

**S4 Table. Primer used for qPCR.**
(TIFF)

**S5 Table. dsRNAs ordered from DRSC.**
(TIFF)

**S1 Text. Supplemental procedures.**
(PDF)

## Acknowledgments

We thank Tristan Naas and Jenny Nguyen for their technical support, Matthieu Bellet for the endosiRNA-sensor constructs. We thank Jean-Luc Imler and Joao Marques for the critical reading of the manuscript, and Alfredo Castello for his scientific advice. The mass spectrometry instrumentation was funded by the University of Strasbourg IdEx "Equipement mi-lourd" 2015 to Plateforme Protéomique Strasbourg-Esplanade.

## Author contributions

**Conceptualization:** Claire Rousseau, Gabrielle Haas, Philippe Hammann, Carine Meignin.

**Data curation:** Claire Rousseau, Lauriane Kuhn, Johana Chicher, Carine Meignin.

**Formal analysis:** Claire Rousseau, Lauriane Kuhn, Johana Chicher, Carine Meignin.

**Funding acquisition:** Carine Meignin.

**Investigation:** Claire Rousseau, Thomas Morand, Gabrielle Haas, Emilie Lauret, Lauriane Kuhn, Philippe Hammann.

**Methodology:** Claire Rousseau, Gabrielle Haas, Johana Chicher, Philippe Hammann.

**Project administration:** Carine Meignin.

**Resources:** Philippe Hammann, Carine Meignin.

**Software:** Claire Rousseau, Carine Meignin.

**Supervision:** Carine Meignin.

**Validation:** Claire Rousseau, Thomas Morand, Gabrielle Haas, Carine Meignin.

**Visualization:** Claire Rousseau, Thomas Morand, Gabrielle Haas, Lauriane Kuhn, Johana Chicher, Carine Meignin.

**Writing – original draft:** Claire Rousseau, Lauriane Kuhn, Johana Chicher, Philippe Hammann, Carine Meignin.

**Writing – review & editing:** Claire Rousseau, Thomas Morand, Gabrielle Haas, Carine Meignin.

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
