## [Decision Letter · Decision Letter 0]

1 May 2024

Dear Dr Meignin,

Thank you very much for submitting your manuscript "In vivo Dicer-2 interactome during viral infection reveals novel pro and antiviral factors in Drosophila melanogaster" for consideration at PLOS Pathogens. As with all papers reviewed by the journal, your manuscript was reviewed by members of the editorial board and by several independent reviewers. In light of the reviews (below this email), we would like to invite the resubmission of a significantly-revised version that takes into account the reviewers' comments.

While the three reviews for your manuscript range in their scope, two of the reviewers suggest that more mechanistic detail is needed to elevate the impact of your manuscript. Specifically, the function of the identified Dicer interactors should be delineated experimentally to support their proviral or antiviral role. Are the interactors' roles in mediating virus replication due to their interaction with Dicer or another independent function?

Additionally, the reviewers have suggested a number of revisions to improve clarity in the text and rationale for experimental design.

We cannot make any decision about publication until we have seen the revised manuscript and your response to the reviewers' comments. Your revised manuscript is also likely to be sent to reviewers for further evaluation.

Sincerely,

Alan G. Goodman

Guest Editor

PLOS Pathogens

Sonja Best

Section Editor

PLOS Pathogens

Michael Malim

Editor-in-Chief

PLOS Pathogens

orcid.org/0000-0002-7699-2064

While the three reviews for your manuscript range in their scope, two of the reviewers suggest that more mechanistic detail is needed to elevate the impact of your manuscript. Specifically, the function of the identified Dicer interactors should be delineated experimentally to support their proviral or antiviral role. Are the interactors' roles in mediating virus replication due to their interaction with Dicer or another independent function?

Additionally, the reviewers have suggested a number of revisions to improve clarity in the text and rationale for experimental design.

Reviewer's Responses to Questions

**Part I - Summary**

Reviewer #1: The research article “In vivo interactome during viral infection reveals novel pro and antiviral factors in Drosophila melanogaster” by Claire Rousseau and colleagues, the authors aim to characterize the interacting partners of Dicer-2 in adult flies and identify the differential composition of the interactome upon infection and in the context of previously characterized Dicer-2 mutants. In general the paper promised to provide a valuable resource that is of broad interest to the insect immunity field. In particular, profiling the interactome of one of the key antiviral factors in vivo and in an infection setting is an approach that should be praised. Also the comparison of distinct mutants in principle has the power to identify protein complexes in association of Dicer-2 that is compromised in various aspects of its physiological function. However, the data analysis as it presented in the current version of the manuscript requires serious revision and some additional experiments should be included to strengthen the conclusions made. Please find below my specific comments.

Reviewer #2: In this manuscript the authors determine the interactome of Drosophila melanogaster Dicer-2, as well as mutants of Dicer-2 with point mutations in the helicase or RNase III domains. The authors note this is the first interactome for Dicer-2 that was performed in vivo in adult flies.

The authors have dicer-2 null flies that are complemented with WT or mutant versions of GFP::Dicer-2, offering a great system for the IP-mass spectrometry protocol. Multiple replicates were included, and interactomes were determined with and without DCV infection, and with and without treatment with RNase to distinguish direct or RNA-mediated interactions.

The study is very well controlled and well performed, and the resulting interactome robust and convincing. The LC-MS/MS analyses showed expected proteins indicating the pipeline was working, as well as those not previously known to interact with Dicer-2.

The authors follow up on proteins of interest, using western analyses, RNAi knockdown in S2 cells, as well as knockdown in whole animals, where survival was monitored. These studies indicate that both eiF4G1 and Rin are involved in antiviral defense, but here and in all cases the study stops short of providing mechanistic information about how these proteins are functioning in viral infection or with Dicer.

That said, the robust datasets will serve as a great resource for future studies by researchers studying Dicer and the antiviral response, as well as those with a general interest in RNA processing.

Reviewer #3: In this manuscript the authors analyzed the Dicer interactome in adult flies using mass spectrometry, in non-infected and virus infected flies. They then go on to perform a functional screen in Drosophila S2 cells and adult flies to analyze whether the interactants affect virus replication. They identified previously known as well as novel Dicer interactants and report amongst others Rasputin (Rin) as factor that may be involved in the antiviral response against DCV. Overall, the work is well designed and well executed, and the data are comprehensively reported. However, functional or mechanistic follow-up on the role of the interactors in Dicer function is lacking from the manuscript. This is unfortunate, as the authors highlight the major open question in their introduction and abstract: it is unknown how Dicer recognizes the protected termini, and it would indeed be of significant interest to analyze whether Dicer cofactors help viral RNA recognition and processing. Therefore, the manuscript remains a hypothesis-generating resource, but does not provide novel insights into viral RNA recognition.

**Part II – Major Issues: Key Experiments Required for Acceptance**

Reviewer #1: • I was puzzled by the choice of negative controls. Why has no ubi>GFP line been used that matches the genetic background of the rescue lines? Is the Canton S fly of the same genetic background as the transgene expressing lines? This was not clear to me and should be better explained in the methods section. If the negative control lines are of a different background, this is not ideal for the analysis. In this case, can the authors comment on the question whether protein expression of distinct genetic backgrounds is comparable, perhaps using published proteomics data, if available.

• The description of mass spec data analysis is hard to follow and should be clarified. It is unclear to me, how the authors distinguish the “overall/global” Dicer-2 analysis from the specific analysis of the wild-type construct and the two mutant constructs. Also in the global analysis shown in Fig 1C and S1C, it is possible to distinguish differential enrichment of interacting partners to different constructs in the mock and virus infected condition. Also the purpose of an overall analysis is not entirely clear to me, given that the introduction of the manuscript highlights the molecular difference between the distinct Dicer-2 constructs.

The authors present the analysis of overall Dicer interactome and the specific analysis of mutants as “two different methods” (line 209) which I find quite a stretch given that the experimental input for these analyses are the same. Due to this double analysis presented in figures 1 and 2 it becomes very hard to follow which protein factors the authors consider general interaction partners and more specific factors that are interacting in a construct/infection dependent status. Given that the authors consider their data a resource to the community, it would be ideal to have an easy-to-interpret representation of the data. I would recommend to analyze the interactome of wild-type and mutant mass spec data separately in uninfected and infected cells (as done for fig 2, I assume). These data should be shown as a figure for all constructs, not only the wild type as an example. Also a supplemental table with raw mass spec, akin to the table of candidates in Suppl. Table 1 data but then for all interactors in each condition would be helpful for the interested reader, who wants to use the data as resource. When the interaction partners in individual conditions are established, overlapping factors can be identified between mutants and wildtype in the uninfected and infected conditions separately. Only hereafter, I would recommend to dive into the interaction of Dicer-2 networks being modulated by DCV infection. Ideally this should be done using a plot that shows the underlying data (for instance a scatter-plot showing correlation between noninfected and DCV infected samples) Venn diagrams by themselves are not particularly informative. The authors may find a different order of analysis more useful, but it is highly recommended to reorder the line of thoughts, as the current representation blurs main messages. I think this is a missed chance.

• The choice of top 15% to define candidates and predict interaction networks is quite random. Well known interactors of Dicer-2 like Loqs are identified but have a lower rank number, suggesting that even a less stringent analysis can possibly identify relevant Dicer-2 interactors. The argument that 36 genes out of the 44 overall Dicer interactors are shared between the 146 proteins common to all constructs IPs is not an independent proof, as the analyses rely on the same data.

• It is odd to calculate an overall enrichment for Dicer interactors upon DCV infection (Fig 2A). Figure S1A shows that an enrichment upon infection can be different for each Dicer-2 construct (e.g. NinaC, Cup, Me31B). By analyzing DCV enrichment as a whole, one loses this important information. As indicated in my previous comment, the interacting partners should first be identified separately in different conditions to take this analysis as a basis for further comparisons.

• The authors should explain their choice for the proteins that are validated by Western blot in Fig 3. Now it feels random. The figure would have more added value, if more of the candidates that are later followed upon had been included (for example: eIF4G, rin, lig)

• The co-IP of the large isoform of Syp with Dicer-2 is not reproduced in the experiment shown in Figure 3B, arguing against a very stable interaction.

• The data on RNase-treated samples is described only very superficially. How many of the earlier hits are reproducible? which ones? Also what is the identify of RNase dependent interactors vs RNase independent interactors. The authors should include a proper description of the data. Just showing a Venn diagram does not provide a lot of information.

• The last sentence of the abstract is an overstatement. The authors have not formally proven that the changed virus levels upon knockdown of selected candidate genes are mediated through the interaction of that protein with Dicer-2 in RNAi (or beyond). Currently, the protein-protein interaction and the antiviral phenotype are two independent observations. Would the antiviral or proviral phenotype of candidates still be visible in Dicer null mutants or can the authors show that the knockdown affects Dicer-2 function (such as siRNA binding, dsRNA cleavage). The authors should show using such genetics or biochemical assays the role of the interactor in Dicer-2 antiviral activity.

Reviewer #2: (No Response)

Reviewer #3: (No Response)

**Part III – Minor Issues: Editorial and Data Presentation Modifications**

Reviewer #1: • It is unclear throughout the manuscript which negative control line has been used to compare the interactome to? The WT or GFP control or an average of the two. The authors should specify this.

• The authors should specify early in the result section that this MS analysis has been performed in triplicate, which strengthens the case of statistical robustness of data.

• Why are some genes depicted as individual isoforms (such as Homer & Homer RB) whereas other gene isoforms are collapsed into a single entry (such as syp). As this may skew data analysis, it is recommended to have a homogenous approach across all genes.

• The rescue phenotypes of the wild type and mutants (shown in Fig S1B) should be explicitly stated rather than referring to the literature (line 133-134)

• The term “early” steps of dsRNA recognition (e.g. line 131 and 198) is misleading as it implies a temporal component of the experiment. Yet, the timeline of the experiment has been identical for all constructs. The idea that the mutant are stuck in processing may not necessarily reflect time and should be rephrased.

• Fig 2B and 2C. What does “fold change” refer to? Spectral count? Relative protein abundance. Please specify

• Fig 4B: doublecheck the Y-axis. Currently the data suggests a n up to 10^10 increase in viral RNA levels, which I can barely believe to be true. Also, is the data normalized against LacZ condition? In that case, please correct in the methods, that you plot the delta delta CT values.

• Fig 5C: I assume the data is shown on a log scale which is not indicated in the y-axis description.

• FigS5: I am surprised about the apparent randomness of GO terms that appear in the wild type versus mutant analysis. Does that mean that the data is rather noisy and is the overall GO analysis shown in Fig1 then not heavily biased by the Helicase mutant interactome?

• Line 268: Where does the number 448 come from, the figure shows 317 proteins

• The authors should discuss the limitation that their assay profiles interactomes in a blend of different tissues from males and females and that interactomes may differ in distinct organs or cell types which cannot be captured with the assay employed in this study.

• I have some suggestions for textual adaptations: I recommend to carefully spell-check the manuscript or let a native speaker proof-read the text.

o Line 126: delete “the dynamics of” the study compares two steady states (infected and non infected) not a dynamic change of interactomes in time or space

o Line 170: help us to better understand

o Line 188 delete “s” in RNAs degradation

o Line 197: replace “be” with “are”

o Line 206: remove “of them”

o Line 213: replace this data with these data

o Line 512: Sentence is not finished

o Line 525 replace “on RNAi” with “an RNAi”

Reviewer #2: 1. The included confirmation of interactions by western blot is an important validation, but would be more convincing if the authors indicated in the legends how reproducible the result was, that is, how many times was each western performed?

2. Given that this paper is focused on Drosophila, in addition to citing the mammalian paper on Arc1, it would be appropriate to cite the sister paper about Drosophila Arc1( Ashley J, Cordy B, Lucia D, Fradkin LG, Budnik V, Thomson T. Retrovirus-like Gag Protein Arc1 Binds RNA and Traffics across Synaptic Boutons. Cell. 2018;172(1-2):262-74 e11. doi: 10.1016/j.cell.2017.12.022. )

Reviewer #3: Suggestions and comments:

- Figure 1B: please define CS

- Figure 1C: The outer circles indicating significance are not clearly visible

- Figure 1E: Please define the line widths and colors

- Figure 2D: why are there two dots for Dicer-2?

- Figure 3: staining for a housekeeping protein would be useful.

- Figure 4: Knockdown of the positive control AGO2 does not affect viral replication, which the authors attribute to using RNAi to knock down an RNAi factor. Yet, such RNAi-of-RNAi approaches have been successfully used before, also in relation to viral infection. Also, the entire screen is to some aspect biased in the same way, RNAi is used to knock down genes that are likely involved in RNAi. The possibility that this would lead to false negatives merits discussion.

- Figure 4: please define the box plot (mean, whiskers, etc) in the legend.

- All figures: please define the statistical test in the legends.

- The prefix sh and ds is not consistently used (shAGO2 and shmCherry)

- Survival curves: please report the number of flies and number of replicate experiments.

- I was surprised to find the strong significant effects in the survival curves, given the that some of them are, in fact, overlapping (e.g., Rm62, Rump, Rin). Please doublecheck the statistics.

PLOS authors have the option to publish the peer review history of their article (what does this mean? ). If published, this will include your full peer review and any attached files.

**Do you want your identity to be public for this peer review?** For information about this choice, including consent withdrawal, please see our Privacy Policy .

Reviewer #1: No

Reviewer #2: No

Reviewer #3: No
---

## [Decision Letter · Decision Letter 1]

26 Jan 2025

PPATHOGENS-D-24-00621R1

In vivo Dicer-2 interactome during viral infection reveals novel pro and antiviral factors in Drosophila melanogaster

PLOS Pathogens

Dear Dr. Meignin,

Thank you for submitting your manuscript to PLOS Pathogens. After careful consideration, we feel that it has merit but does not fully meet PLOS Pathogens's publication criteria as it currently stands. Therefore, we invite you to submit a revised version of the manuscript that addresses the points raised during the review process.

Please submit your revised manuscript within 60 days Mar 27 2025 11:59PM. If you will need more time than this to complete your revisions, please reply to this message or contact the journal office at plospathogens@plos.org. Please include the following items when submitting your revised manuscript:

We look forward to receiving your revised manuscript.

Kind regards,

Alan G. Goodman

Academic Editor

PLOS Pathogens

Sonja Best

Section Editor

PLOS Pathogens

 Sumita Bhaduri-McIntosh

Editor-in-Chief

PLOS Pathogens

orcid.org/0000-0003-2946-9497

Michael Malim

Editor-in-Chief

PLOS Pathogens

orcid.org/0000-0002-7699-2064

**Additional Editor Comments :**

All reviewers agree that your revised manuscript has been greatly improved. In general, they ask for minor revisions to the text and one additional experiment. Specifically, they request sequencing of small RNAs (viral and endogenous) as a readout of Dicer2 activity upon knockdown of the Dicer2 interactors. These data would bolster the virus replication data following Dicer2 interactor knockdown that is already included in the manuscript.

**Journal Requirements:**

1) We note that your "3_Dicer2_2402_final_text." files are duplicated on your submission. Please remove any unnecessary or old files from your revision, and make sure that only those relevant to the current version of the manuscript are included.

- ® on pages: 40 line 1153, 41 lines 1202, and 42 line 1220.

- TM on pages: 41 line 1203, 42 lines 1223, 1227, 1228, 1233, and and 1242.

3) We notice that your supplementary information (Supplemental Procedures) is included in the manuscript file. Please remove them from the main file. Please ensure that each Supporting Information file has a legend listed in the manuscript after the references list.

Potential Copyright Issues:

i) Figure 6A. Please confirm whether you drew the images / clip-art within the figure panels by hand. If you did not draw the images, please provide (a) a link to the source of the images or icons and their license / terms of use; or (b) written permission from the copyright holder to publish the images or icons under our CC BY 4.0 license. Alternatively, you may replace the images with open source alternatives. See these open source resources you may use to replace images / clip-art:

ii)  Figures 1B and 5A: Please provide proper attributions to the sources in the legends of the figures.

5) Thank you for providing us with your Data Availability statement.  Please note that, though access restrictions are acceptable now, your entire minimal dataset will need to be made freely accessible if your manuscript is accepted for publication. This policy applies to all data except where public deposition would breach compliance with the protocol approved by your research ethics board. If you are unable to adhere to our open data policy, please kindly revise your statement to explain your reasoning and we will seek the editor's input on an exemption.

7) Please ensure that the funders and grant numbers match between the Financial Disclosure field and the Funding Information tab in your submission form. Note that the funders must be provided in the same order in both places as well.

Please indicate by return email the full and correct funding information for your study and confirm the order in which funding contributions should appear. Please be sure to indicate whether the funders played any role in the study design, data collection and analysis, decision to publish, or preparation of the manuscript.

**Reviewers' Comments:**

Reviewer's Responses to Questions

**Part I - Summary**

Reviewer #1: I would like to compliment the authors for this thorough revision. Readibility and coherence has improved significantly. I appreciate the effort that has been put to address many of the reviers' concerns. I only have a few minor points of feedback.

Reviewer #2: In this revised manuscript the authors have responded to prior criticisms primarily by changing the way they analyzed their mass spectrometry data, and including additional transgenic flies expressing GFP alone, to evaluate non-specific interactions. While nicely increasing confidence in the identified interacting proteins, for the most part the factors identified remain the same.

As in the prior submission, the authors include experiments to validate the observed interactions, using immunoprecipitation experiments and western analyses, and they also perform knockdown experiments to evaluate effects on viral load and survival. The latter experiments set the stage for further studies to understand the mechanism by which these factors affect antiviral defense, albeit such studies are not included in this publication.

Regardless, it is still my opinion that the robust datasets will serve as a great resource for future studies by researchers studying Dicer and the antiviral response, as well as those with a general interest in RNA processing.

Reviewer #3: In this revised manuscript, the authors analyzed the Dicer interactome in adult flies using mass spectrometry, in non-infected and virus infected flies. As already indicated in my original review, the work is well designed and well executed, and the data are comprehensively reported. The paper provides interesting leads to follow-up studies of Dicer-2 interactors.

In my review of the original submission, I expressed doubts about the appropriateness for PLOS Pathogens given the lack of mechanistic insight. My assessment still applies; interesting questions raised in the introduction are not addressed in the manuscript. However, as the authors were given the opportunity to revise their manuscript and since they have extensively revised it and addressed my comments, I now support publication in PLOS Pathogens. There is, however, one essential experiment that needs to be done.

**Part II – Major Issues: Key Experiments Required for Acceptance**

Reviewer #1: none

Reviewer #2: (No Response)

Reviewer #3: The effect of the interactors on Dicer-2 enzymatic activity has not analyzed. An analyses of siRNA profiles (viral and endogenous) upon knockdown of key interactors would greatly improve the value of the manuscript as it is the most direct readout of Dicer-2 function.

In addition, the results of the endosiRNA reporter have not been discussed in the main text (except in the context that Ago2 knockdown affects the endoRNAi reporter). A notable result from that experiment is that none of the interactors affect endoRNAi activity, suggesting that they do not affect siRNA production. This merits to be discussed in the Results section.

**Part III – Minor Issues: Editorial and Data Presentation Modifications**

Reviewer #1: I find the MDS analysis in response to one of my earlier questions a rather useful piece of data to obtain a first impression of overall trends in the data. I would highly recommend to include this panel in supplemental figure 1, alongside a brief description in the main text.

Line 263. Change to …, and the predicted interactions between those candidates was represented…

Not all links in a STRING network are truly validated physical protein-protein interactions.

At various points the authors refer to the S2 cell experiments as “ex vivo” which is a term referring to tissue cultures explanted from living organisms. I propose to change this to “in cells”.

The lack of formal experimental evidence if identified interaction partners act on virus replication through RNAi is worth mentioning in the discussion.

Reviewer #2: (No Response)

Reviewer #3: Line 149: I think it should suffice to mention numbers of replicates in legends, not in the main text.

Line 180: With reference to Veneno, the authors may also want to discuss the Brosh et al paper (PMID: 35858337), showing an antiviral phenotype of a truncated Veneno allele.

Line 199: a case is made that Lost is especially enriched in Helicase mutant. However, this conclusion is based on the p value, but the difference in Lost enrichment seems to be minor based on the fold enrichment (which is more informative than the p value). I suggest rephrasing.

Line 227-228: “these interactants include the viral RNA-dependent RNA polymerase (RdRp) from DCV (ORF1)”. I found this phrase a bit ambiguous: the reference to ORF1, which encodes also other viral proteins, suggests that there may be other viral Dicer-2 interactors in addition to the RdRP. Please clarify, or delete the reference to ORF1 in the sentence.

Figure 3, please define BFDR

IP mass spec was done on lysates of total animals, but I expect that only a minority of the cells in the entire animal is DCV infected. Hence the majority of cells in the infected samples may in fact be noninfected cells. Could the authors comment on this? How was the inoculum and time of analysis chosen? Did the authors monitor viral loads / number of infected cells at the indicated time point?

PLOS authors have the option to publish the peer review history of their article (what does this mean? ). If published, this will include your full peer review and any attached files.

**Do you want your identity to be public for this peer review?** For information about this choice, including consent withdrawal, please see our Privacy Policy .

Reviewer #1: No

Reviewer #2: No

Reviewer #3: No

**Figure resubmission:**
---

## [Editor Report · Decision Letter 2]

1 Apr 2025

Dear Dr Meignin,

We are pleased to inform you that your manuscript 'In vivo Dicer-2 interactome during viral infection reveals novel pro and antiviral factors in Drosophila melanogaster' has been provisionally accepted for publication in PLOS Pathogens.

Best regards,

Alan G. Goodman

Academic Editor

PLOS Pathogens

Sonja Best

Section Editor

PLOS Pathogens

Sumita Bhaduri-McIntosh

Editor-in-Chief

PLOS Pathogens

orcid.org/0000-0003-2946-9497

Michael Malim

Editor-in-Chief

PLOS Pathogens

orcid.org/0000-0002-7699-2064

Thank you for the revision of your manuscript that addressed the remaining concerns of the reviewers.
---

## [Editor Report · Acceptance letter]

Dear Dr Meignin,

We are delighted to inform you that your manuscript, "*In vivo* Dicer-2 interactome during viral infection reveals novel pro and antiviral factors in *Drosophila melanogaster* ," has been formally accepted for publication in PLOS Pathogens.

Best regards,

Sumita Bhaduri-McIntosh

Editor-in-Chief

PLOS Pathogens

orcid.org/0000-0003-2946-9497

Michael Malim

Editor-in-Chief

PLOS Pathogens

orcid.org/0000-0002-7699-2064